# The fitness landscape of the African *Salmonella* Typhimurium ST313 strain D23580 reveals unique properties of the pBT1 plasmid

Rocío Canals[1]*, Roy R. Chaudhuri[2], Rebecca E. Steiner[3,4], Siân V. Owen[1¤], Natalia Quinones-Olvera[5], Melita A. Gordon[6,7], Michael Baym[5], Michael Ibba[3,4], Jay C. D. Hinton[1]

**1** Institute of Integrative Biology, University of Liverpool, Liverpool, United Kingdom, **2** Department of Molecular Biology and Biotechnology, University of Sheffield, Sheffield, United Kingdom, **3** Department of Microbiology, The Ohio State University, Columbus, Ohio, United States of America, **4** Center for RNA Biology, The Ohio State University, Columbus, Ohio, United States of America, **5** Department of Biomedical Informatics, Harvard Medical School, Boston, Massachusetts, United States of America, **6** Institute of Infection and Global Health, University of Liverpool, Liverpool, United Kingdom, **7** Malawi-Liverpool-Wellcome Trust Clinical Research Programme, University of Malawi College of Medicine, Blantyre, Malawi, Central Africa

¤ Current address: Department of Biomedical Informatics, Harvard Medical School, Boston, Massachusetts, United States of America

* rcanals@gmail.com

**Data Availability Statement:** Illumina sequencing data have been deposited in European Nucleotide Archive (ENA) repository (EMBL-EBI) under

## Abstract

We have used a transposon insertion sequencing (TIS) approach to establish the fitness landscape of the African *Salmonella enterica* serovar Typhimurium ST313 strain D23580, to complement our previous comparative genomic and functional transcriptomic studies. We used a genome-wide transposon library with insertions every 10 nucleotides to identify genes required for survival and growth *in vitro* and during infection of murine macrophages. The analysis revealed genomic regions important for fitness under two *in vitro* growth conditions. Overall, 724 coding genes were required for optimal growth in LB medium, and 851 coding genes were required for growth in SPI-2-inducing minimal medium. These findings were consistent with the essentiality analyses of other *S.* Typhimurium ST19 and *S.* Typhi strains. The global mutagenesis approach also identified 60 sRNAs and 413 intergenic regions required for growth in at least one *in vitro* growth condition. By infecting murine macrophages with the transposon library, we identified 68 genes that were required for intra-macrophage replication but did not impact fitness *in vitro*. None of these genes were unique to *S.* Typhimurium D23580, consistent with a high conservation of gene function between *S.* Typhimurium ST313 and ST19 and suggesting that novel virulence factors are not involved in the interaction of strain D23580 with murine macrophages. We discovered that transposon insertions rarely occurred in many pBT1 plasmid-encoded genes (36), compared with genes carried by the pSLT-BT virulence plasmid and other bacterial plasmids. The key essential protein encoded by pBT1 is a cysteinyl-tRNA synthetase, and our enzymological analysis revealed that the plasmid-encoded CysRS$^{pBT1}$ had a lower ability to charge tRNA than the chromosomally-encoded CysRS$^{chr}$ enzyme. The presence of aminoacyl-tRNA

accession number PRJEB33138. Dalliance website is available at the URL: https://hactar.shef.ac.uk/D23580.

**Funding:** This work was supported by a Wellcome Trust Senior Investigator award to JCDH (Grant 106914/Z/15/Z), and a National Institutes of Health grant to MI (GM065183). RC was supported by a EU Marie Curie International Incoming Fellowship (FP7-PEOPLE-2013-IIF, Project Reference 628450). The funders had no role in study design, data collection and analysis, decision to publish, or preparation of the manuscript.

**Competing interests:** The authors have declared that no competing interests exist.

synthetases in plasmids from a range of Gram-negative and Gram-positive bacteria suggests that plasmid-encoded essential genes are more common than had been appreciated.

## Author summary

The success of a bacterial pathogen requires a trade-off between fitness and the need to produce the energy-demanding virulence systems that promote survival and growth in an infected host. Not only must the pathogen express the right virulence genes in the right place at the right time, but the repertoire of genes within an individual bacterial isolate must be optimised to maintain fitness during competition with other bacteria. Here, we identified the genetic requirement of *S.* Typhimurium ST313 strain D23580 for survival in different *in vitro* conditions and for replication inside mammalian macrophages. Our comparative analyses reveal that the function of housekeeping and virulence genes is highly conserved between *S.* Typhimurium ST313 and ST19. One of the D23580 plasmids, pBT1, carried an unprecedented low number of transposon insertions across 38% of the genes compared to the other large D23580 plasmid, pSLT-BT, and other reported transposon mutagenesis studies in plasmids. The essential pBT1-encoded cysteinyl-tRNA synthetase CysRS$^{pBT1}$ showed a lower enzymatic activity and reduced stability compared with the chromosomally-encoded CysRS$^{chr}$. Because plasmid-encoded aminoacyl-tRNA synthetases are found in a range of bacteria, we propose that the role of essential genes carried by plasmids for determining microbial fitness deserves further study.

## Introduction

*Salmonella* spp. are important pathogens of humans and animals. In humans, salmonellosis is classified as either a typhoidal or non-typhoidal *Salmonella* (NTS) disease. Typhoidal salmonellosis involves systemic spread through the body that causes enteric fever, and is associated with the *S. enterica* serovars Typhi (*S.* Typhi) and Paratyphi (*S.* Paratyphi). In contrast, NTS disease normally involves a self-limiting gastroenteritis that is transmitted via food, involving approximately 94 million human cases and about 155,000 deaths [1]. The *S. enterica* serovar Typhimurium (*S.* Typhimurium) sequence type ST19 causes the majority of gastroenteritis in immuno-competent individuals worldwide via pathogenic mechanisms that induce mucosal inflammatory responses in the gut. *S.* Typhimurium can thrive in this inflamed gut whilst other key members of the gut microbiota cannot survive [2,3]. The remarkable ability of this pathovariant to enter, survive, and proliferate in mammalian macrophages and epithelial cells in a "*Salmonella*-containing vacuole" (SCV) is responsible for systemic disease in both animals and humans [4].

The HIV epidemic in sub-Saharan Africa has been implicated in the evolution of new clades of NTS strains that cause bacteraemia in humans. Specifically, the HIV virus impairs the immunity of adults, a phenomenon that occurred concurrently with the development of NTS strains able to cause a systemic disease, invasive non-typhoidal salmonellosis (iNTS) [5–9]. In children, malaria and malnutrition are also risk factors for iNTS [10]. The *S.* Typhimurium and *S.* Enteritidis isolates responsible for invasive NTS isolates have a multi-drug-resistant phenotype, necessitating the replacement of conventional therapies with alternative antibiotics [6,11,12].

In sub-Saharan Africa, *S.* Typhimurium strains belonging to sequence type ST313 have been associated with the majority of systemic disease, causing hundreds of thousands of deaths in 2010 [13]. The genome sequence of one representative of ST313, D23580, was published in 2009 [14], and was recently updated [15].

To date, genome-wide functional genomic studies have focused on the fitness of *S.* Typhimurium and *S.* Typhi in several *in vitro* growth conditions, and within eukaryotic cells and animal infection models [16,17]. The recent development of transposon-insertion sequencing (TIS) technology combines global mutagenesis and high-throughput sequencing to functionally characterize bacterial genes. Transposon insertion libraries are constructed in a strain of interest, in which nonessential genes for a particular growth condition (input library) are dispensable and contain insertions. This library of random transposon insertion mutants can be used to identify genes "required" for fitness under that particular environmental condition (output library). The relative proportion of each mutant in the input and the output libraries is determined by high-throughput sequencing, a strategy that enables the fitness contribution of each gene to be quantified in environmental conditions of interest [18,19]. The first study that used this technology in *Salmonella* described a new TIS strategy: Transposon-Directed Insertion Site Sequencing (TraDIS) [20]. Subsequently, various TIS-based strategies have been used for functional genomic analysis of *Salmonella* serovars Typhimurium and Typhi [21–23].

Here we report the TIS-based identification of the genes of *S.* Typhimurium ST313 D23580 responsible for growth and survival inside murine macrophages, and the genetic requirements of this strain to grow and survive in laboratory conditions (summarized in Fig 1).

## Results and discussion

### Transposon insertion profile of a *S.* Typhimurium D23580 Tn5 library

A transposon library was constructed in the *S.* Typhimurium ST313 lineage 2 representative strain D23580. The pool of transposon mutants was grown in LB (input) and successively passaged three times in two different laboratory growth media: a rich medium, LB (output); and an acidic phosphate-limiting minimal medium that induces *Salmonella* pathogenicity island (SPI) 2 expression, designated InSPI2 (Fig 1). Genomic DNA from the input and output samples was purified and prepared for Illumina sequencing of the DNA adjacent to the transposons (Materials and Methods). S1 Table shows the number of sequence reads obtained, the sequence reads that contained the transposon tag sequence, and the sequence reads that were uniquely mapped to the *S.* Typhimurium D23580 genome.

Sequence analysis of the input sample identified 797,000 unique transposon insertion sites in *S.* Typhimurium D23580, equating to an average of one transposon integration every six nucleotides. All data are available for visualization in a Dalliance genome browser [24] which shows the transposon insertion profile of the chromosome and the four plasmids (pSLT-BT, pBT1, pBT2, pBT3) in *S.* Typhimurium D23580: https://hactar.shef.ac.uk/D23580. The number of reads, transposon insertion sites, insertion index, and "requirement" call per gene are summarized in S2 Table. The insertion index was calculated as described in Materials and Methods and allowed the genetic requirements of *S.* Typhimurium D23580 for growth to be determined after a single passage in LB. The genes designated "required" included essential genes and genes that contributed to fitness in this particular environmental condition. Some genes were called "ambiguous" when they could not be robustly assigned as either required or dispensable by the analysis (Materials and Methods). A total of 596 genes were required in the *S.* Typhimurium D23580 genome: 558 were located in the chromosome, two in the pSLT-BT plasmid, and 36 in the pBT1 plasmid (Fig 2A and 2B).

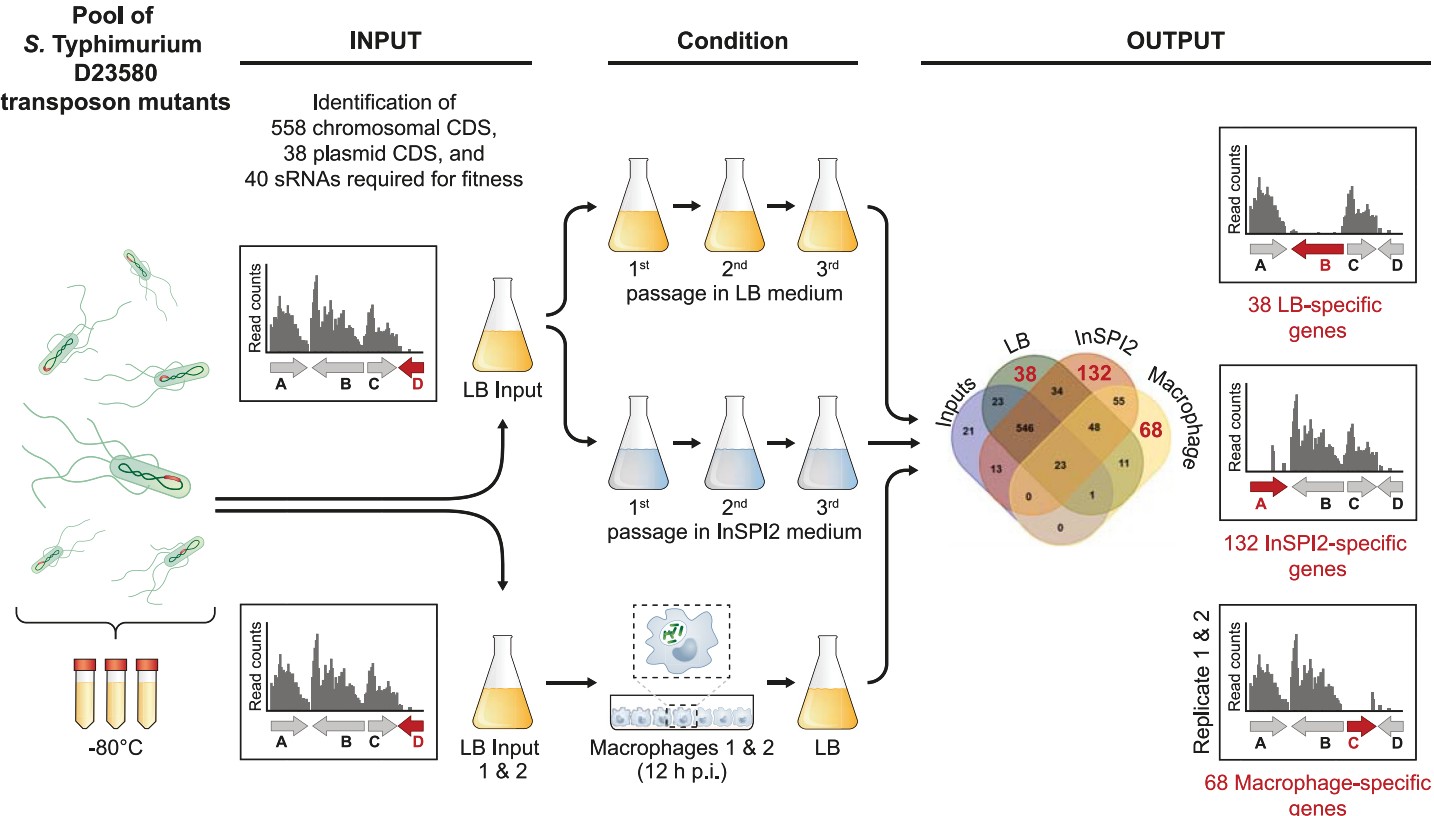

**Fig 1. Transposon-insertion sequencing (TIS) in *S.* Typhimurium ST313 D23580.** Schematic representation of the *S.* Typhimurium D23580 transposon library and growth conditions used in this study. The Venn diagram corresponds to S4B Fig.

To establish common themes amongst closely-related bacteria, the *S.* Typhimurium D23580 chromosomal genes that were required for growth were compared with the genetic requirements of other strains of *S.* Typhimurium (14028, a derivative of SL1344 called SL3261, and LT2) and *S.* Typhi (Ty2, and a derivative of Ty2 named WT174) (Fig 3A, S3 Table) [20,21,25–27]. After the comparison with these five other *Salmonella* isolates, we found that a total of 101 genes were only required in D23580, including one D23580-specific gene encoding the CI^BTP5 repressor of the BTP5 prophage (*STMMW_32121*) [28]. To add context, a Clusters of Orthologous Groups (COG) analysis identified 32 genes that were predominantly assigned to two functional categories of transcription (nine genes) and amino acid transport and metabolism (six genes). Additionally, at least 21 of the 101 genes were associated with virulence: 16 were located in SPI regions, and five encoded associated effectors that were located elsewhere in the genome.

To identify chromosomal genes that are required for growth in *S.* Typhimurium D23580 and other *S.* Typhimurium pathovariants, a comparison with the individual strains was performed (S1A Fig, S4 Table) [21,25]. A total of 250 genes were required in all *S.* Typhimurium and *S.* Typhi strains [20,21,25]. While searching for serovar-specific required genes, we found six genes that were only required by D23580 and the two *S.* Typhimurium strains 14028 and SL3261, but not by *S.* Typhi, namely: *ssaT*, a SPI-2 gene; *STMMW_16291*, encoding a putative amino acid transporter; *hnr*, encoding a regulator; *pth*, encoding a peptidyl-tRNA hydrolase; *STMMW_18451*, with unknown function; and *ddhB* (*rfbG*), an O-antigen gene involved in the biosynthesis of CDP-abequose. Intriguingly, *hns* was only required in *S.* Typhimurium

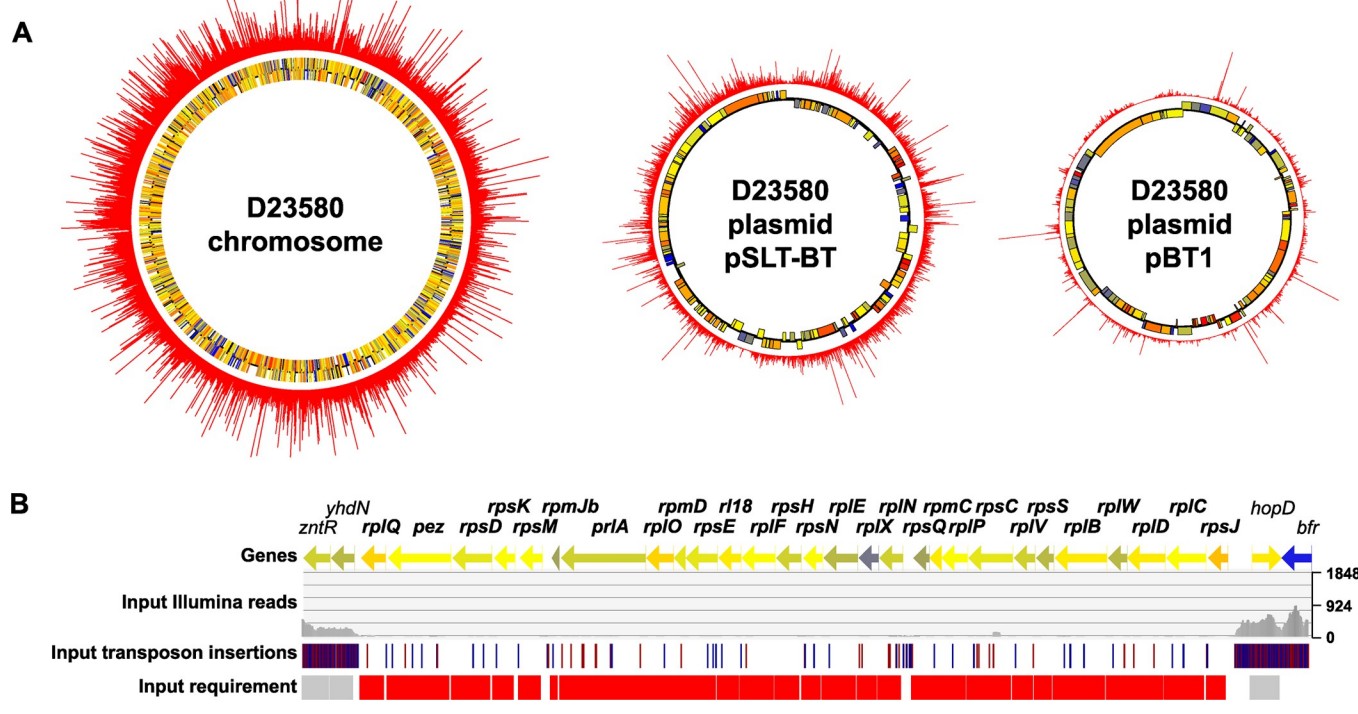

**Fig 2. Transposon insertion profile in *S.* Typhimurium D23580.** (A) Transposon insertion indexes are represented in the outer ring of the chromosome and the pSLT-BT and pBT1 plasmids of *S.* Typhimurium D23580. The two inner rings represent annotated genes coloured according to their GC content (blue = low, yellow = intermediate, red = high). (B) Chromosomal region showing a cluster of genes, between *zntR-yhdN* and *hopD-bfr*, that are required for growth in D23580.

D23580 and 14028 but not in SL3261. In contrast, no genes that were only required by *S.* Typhimurium D23580 and the two *S.* Typhi strains Ty2 and WT174 were found. We conclude that there is a high level of conservation of genes that contribute to fitness during *in vitro* growth of *S.* Typhimurium and *S.* Typhi. Because D23580 is much more closely related to other strains of *S.* Typhimurium than to *S.* Typhi, it is not surprising to see that there is a greater overlap of required genes between the Typhimurium strains than with the Typhi isolates.

Seven genes previously reported to be required in *S.* Typhimurium were not included in the D23580 gene requirements (S1B Fig, S4 Table), prompting a more detailed investigation. Analysis of two other input samples described later in this work showed that two of these genes were consistently identified as dispensable in D23580: *cysS*, encoding a cysteinyl-tRNA synthetase; and *folA*, involved in biosynthesis of tetrahydrofolate. There were 20 genes that had been previously reported in TIS studies to be required in *S.* Typhi but not in D23580 (S1C Fig, S4 Table) [20,21]. Seven out of the 20 genes were consistently found to be dispensable in the two other D23580 input samples analyzed later in this work, and one of them was *cysS*. The cysteinyl-tRNA synthetase is essential for bacterial growth [29]. The dispensability of *cysS* (*cysS^chr^*) in D23580 reflects the fact that the pBT1 plasmid of *S.* Typhimurium D23580 carries the paralogous gene *cysS^pBT1^*, and we published the transposon insertion profiles of these two genes previously [15]. In summary, only the plasmid copy *cysS^pBT1^* was required for growth in LB in D23580, whereas *cysS^chr^* was dispensable.

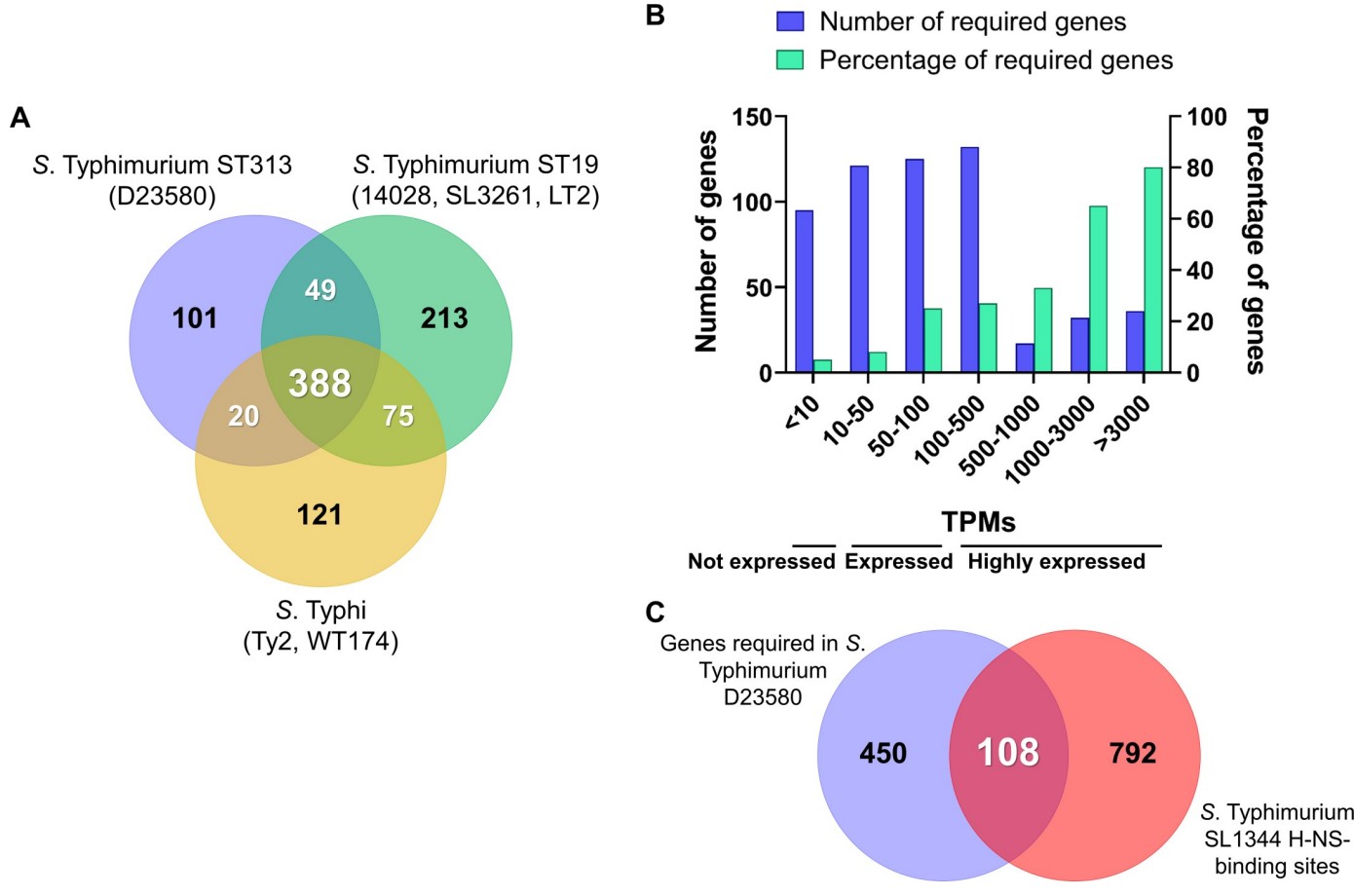

**Fig 3. *S*. Typhimurium D23580 chromosomal genes required for growth *in vitro*.** (A) *S*. Typhimurium D23580 required chromosomal genes for growth that have never been previously identified as required for growth in *S*. Typhimurium (*S*. Typhimurium 14028 [21,26], SL3261 [25], LT2 [27]), or *S*. Typhi (Ty2 [21], WT174 [20]). For the previously published studies, only genes that shared an ortholog in D23580 were used for the final analysis. (B) Transcriptional levels of expression in LB mid-exponential phase (MEP) of the D23580 required genes for growth (data extracted from Canals and colleagues [15]). The X axis represents groups of absolute TPM values. The Y axis represents the number of required genes in D23580. The second Y axis represents the percentage of required genes with a TPM value within the range showed in the X axis out of the total number of genes showing a TPM value within the same range. (C) Required genes in D23580 with H-NS-binding sites reported in *S*. Typhimurium SL1344 [38]. Only the SL1344 genes that shared an ortholog with D23580 were included for the analysis.

## Certain dispensable genes were designated required in the essentiality analysis

Several genes required for *in vitro* growth of D23580 have previously been reported to be dispensable for growth in other *Salmonella* strains in laboratory conditions, including 12 genes located in the SPI-2 pathogenicity island and six genes in the O-antigen biosynthetic cluster [20,21,25]. We considered whether the low number or absence of transposon insertions in certain genes could reflect a limitation of the TIS technique, rather than a strict growth requirement. Although a previous study using a similar strategy for Tn5 transposon library construction did not find bias in the insertion sites [21], a preference of Tn5 for G/C pairs in the target sequence has been reported [30], and motifs for preferential Tn5 integration have been investigated previously [21,31]. Examples of transposon insertion bias include the reduced level of insertions of the Mu element [32,33] or Tn5 and Tn10 transposons [34] in highly transcribed regions of the *Escherichia coli* and *S*. Typhimurium chromosomes.

To correlate the level of transcription with the number of Tn5 transposon insertions, we assessed the absolute expression values of the required genes using our published *S.* Typhimurium D23580 RNA-seq data, grown in LB to mid-exponential phase (MEP) (Fig 3B, S3 Table) [15]. In this dataset, the level of expression of each gene is expressed as Transcripts Per Million (TPM). The majority of required genes showed low and intermediate expression levels, with TPM values between <10 and 500. We found that 80% of the most highly expressed genes (TPM >3000) were required for growth. Of these, 86% were involved in translation, ribosomal structure and biogenesis. We conclude that fewer Tn5 transposon insertions occurred in highly expressed genes and most of these genes are involved in functions that are essential for bacterial growth.

The histone-like nucleoid structuring (H-NS) protein, encoded by the *hns* gene, preferentially binds to A/T-rich regions in bacterial genomes [35,36]. It has been proposed that H-NS-bound DNA could be protected from being a target of transposition and so receive fewer transposon integrations [37]. To investigate whether H-NS-binding explained the low number of transposons found in *Salmonella* genes that are dispensable for growth in laboratory conditions, the reported H-NS-binding sites of *S.* Typhimurium SL1344 [38] were cross-referenced with the list of required genes in D23580 (Fig 3C, S3 Table). A total of 108 genes designated as required in D23580 also contained an H-NS binding site in SL1344 (S3 Table), including 36 genes located in SPI regions and associated effectors. We conclude that only a minority of *S.* Typhimurium D23580 genes are likely to be protected from transposition by H-NS, as discussed below.

Transposon insertions were seen more rarely in SPI pathogenicity island-related *S.* Typhimurium D23580 genes than other parts of the chromosome. Specifically, the *hilC* (SPI-1) and *ssrA* (SPI-2) genes were designated as required in our study. To investigate whether the deletion of these genes imposed a fitness cost upon D23580, we compared the growth of the individual D23580 deletion mutants with the D23580 wild-type (WT) strain in LB (S2A Fig, S4 and S5 Tables). Similar mutants that retained the kanamycin (Km) resistance cassette were also examined, in case the strong promoter of the *aph* gene played a role. The deletion of *ssrA* included the removal of *ssrB*, the two genes that encode the two-component regulatory system of SPI-2. Two mutants that lacked genes involved in the biosynthesis of the lipopolysaccharide (LPS), in *waaL* (lack of O-antigen) and *waaG* (absence of O-antigen and part of the LPS core), were also investigated as examples of genes containing H-NS-binding sites and allowing a high proportion of transposon insertions.

No significant differences were observed in the growth rate of any of the SPI-2 and the SPI-1-defective mutants compared to the WT strain. We observed that the SPI-1 mutant grew to a slightly greater culture density than the WT strain (OD$_{600}$ for the WT was 4.55, and was 4.86 for D23580 Δ*hilC::frt*). This small fitness cost of expressing *hilC* and other SPI-1 genes has already been reported in *S.* Typhimurium, explaining why SPI-1 mutants outcompete the WT strain [39,40]. Both LPS mutants grew slower in LB compared to WT, consistent with previous findings concerning the deletion of the *S.* Typhimurium *waaL* gene [41]. These results suggest that the binding of the H-NS protein to the SPI-1 and SPI-2 regions could explain the low number of transposons in these regions, as no fitness cost for growth in LB of the respective mutants was detected. The high number of transposon insertions found in the LPS genes, which have been reported to contain H-NS-binding sites in *S.* Typhimurium SL1344, and the fact that mutations in these genes had an effect on fitness, indicated that a low number or absence of transposon insertions do not always correlate with the presence of H-NS.

## Genetic requirements for *S.* Typhimurium D23580 growing in rich and SPI-2-inducing media

To build on our previous identification of genes required for survival after a single passage in LB, we studied *in vitro* fitness during growth in nutrient rich and minimal media by comparing the pools of transposon mutants recovered after further passages of the D23580 transposon library in LB (designated as "output"), and the acidic phosphate-limiting minimal medium (PCN, phosphate carbon nitrogen) that induces SPI-2 expression (InSPI2) in *S. enterica*. These data were used to assign an insertion index to each gene (S2 Table).

After three passages in LB, 724 genes were required, which included essential genes and genes that contributed to fitness for *in vitro* growth: 683 genes in the chromosome, 2 genes in the pSLT-BT plasmid, and 39 genes in the pBT1 plasmid. A total of 851 genes were required for optimal growth after three passages in InSPI2: 816 genes in the chromosome, 2 genes in the pSLT-BT plasmid, and 33 genes in the pBT1 plasmid. Genes that had previously been found to be indispensable for growth in the input sample in this study were removed from the lists of required genes (Fig 4A, S3 Table).

There were 54 *S.* Typhimurium D23580 genes that were required for growth after three passages in LB, but not in InSPI2, including the pBT1-encoded *pBT1-0401* and *pBT1-0781*. The gene list included two genes reported to be required by *S.* Typhimurium 14028 after three passages in LB [22], namely *sdhA*, encoding a succinate dehydrogenase flavoprotein subunit; and *sapG* (*trkA*), encoding a potassium transporter. The identification of these particular *S.* Typhimurium genes in multiple TIS studies highlights the importance of *sdhA* and *sapG* for fitness in this environment. The genes *sdhCD*, involved in the conversion of succinate to fumarate (with *sdhAB*), and *fumA*, involved in the conversion of fumarate to malate (with *fumBC*), were also required after three passages in LB in our *S.* Typhimurium D23580 study. Overall, most of the required genes were either involved in energy production and conversion (17%), carbohydrate transport and metabolism (9%), or inorganic iron transport and metabolism (9%).

A total of 191 genes were needed for optimal growth of *S.* Typhimurium D23580 after three passages in the InSPI2 minimal medium. These genes included: *dksA*, that is required for growth on minimal medium [42]; *hfq*, encoding an RNA chaperone that facilitates base-pairing of ~100 small RNAs (sRNAs) to the their target mRNAs [43,44]; and genes involved in thiamine, coenzyme A, biotin, and LPS biosynthesis. The majority of the genes were included in

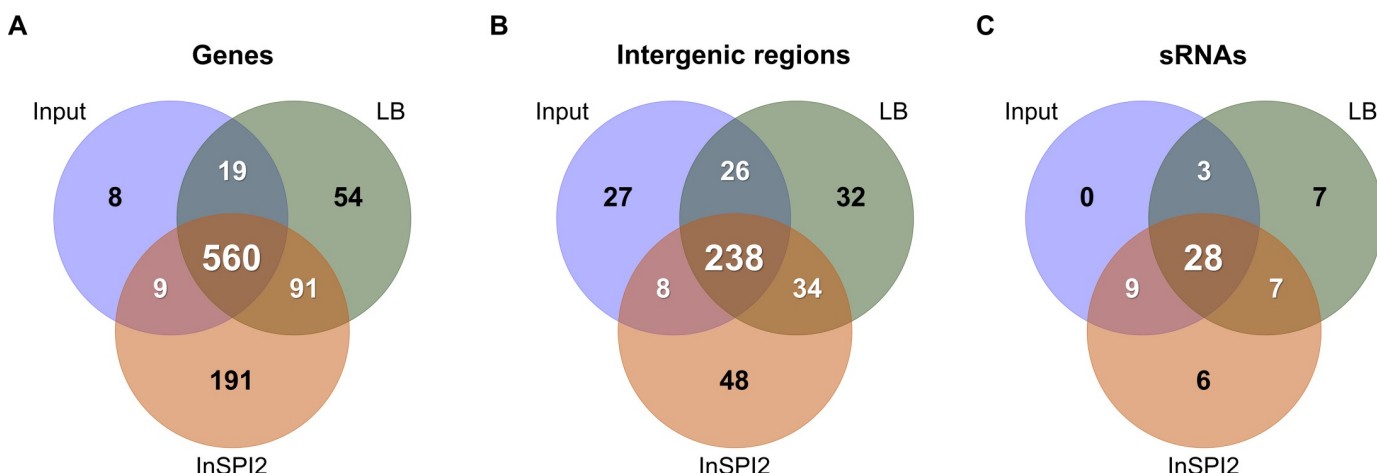

**Fig 4. Identification of *S.* Typhimurium D23580 required for *in vitro* growth.** The figures represent (A) coding genes; (B) intergenic regions; and (C) sRNAs.

the following functional categories: amino acid transport and metabolism (27%), cell envelope biogenesis and outer membrane (7%), inorganic iron transport and metabolism (7%), and nucleotide metabolism (7%).

There were 91 genes designated required in both LB and InSPI2 (Fig 4A, S3 Table). Among them, we found two D23580-specific genes: *pBT1-0081*, in the pBT1 plasmid; and *cI*$^{BTP1}$, encoding the BTP1 prophage repressor [28]. Most of the genes encoded products involved in energy production and conversion (22%), oxidative phosphorylation (12%), and translation, ribosomal structure and biogenesis (10%). These genes reflect the biological needs of *S*. Typhimurium D23580 for growth in laboratory conditions.

## The intergenic regions and sRNAs that increase fitness of *S.* Typhimurium D23580

The high level of saturation of the transposon library allowed the transposon insertion profiles of sRNAs and intergenic regions of ≥100 bp in the chromosome, and the pSLT-BT and pBT1 plasmids, to be investigated to identify the role of these regions in fitness [14,15]. The pBT2 (~2.5 kb) and pBT3 (~2 kb) plasmids were considered as intergenic regions due to the absence of annotation [15]. An insertion index was calculated for each intergenic region and sRNA in the input and LB and InSPI2 output samples as previously described (S6 Table).

A total of 286 intergenic chromosomal regions, 13 intergenic plasmid regions (all in pBT1), and 40 sRNAs were required in the input sample, being essential or contributing to fitness. Thirty-two intergenic regions were important for fitness after three passages in LB, while 48 were only required for growth after three passages in InSPI2 (Fig 4B, S3 Table). Additionally, 34 intergenic regions increased fitness in both LB and InSPI2. Most of the adjacent genes of the intergenic regions that were important for fitness were involved in translation, ribosomal structure and biogenesis (11%), and energy production and conversion (9%). The coding genes required for optimal growth after three passages in LB and InSPI2 also belonged to these two functional categories (S2 Table). The fitness defects of mutants carrying transposon insertions in intergenic regions might be due to disruption of the promoter region of one of the flanking genes, or could reflect mutation of unannotated coding regions.

Seven sRNAs were only required for growth after three passages in LB, but not in InSPI2 minimal media (Fig 4C, S3 Table). In contrast, six sRNAs were identified as being important for fitness after three passages in InSPI2. In total, seven sRNAs that enhanced fitness in both LB and InSPI2 were identified, namely tp2, STnc2010, RyjB, STnc2030, SdsR, STnc3080, and SraG. Among them, SdsR is widely conserved in enterobacteria [45], targeting important global regulators with biological relevance in stationary phase and stress conditions [46] and shown to be required for fitness of *S*. Typhimurium in stationary phase [47]. The fact that inactivation of SdsR has already been described to have a fitness cost in stationary phase helps to validate our TIS approach. For the future, the regulatory targets of the other six sRNAs that are required for growth in both nutrient and minimal media should be determined.

## Intra-macrophage infection with the transposon library suggests the absence of novel virulence factors in *S.* Typhimurium D23580

To build upon our understanding of the *S*. Typhimurium D23580 genes that were required for fitness during growth in nutrient or minimal media, we used the transposon library to investigate the process of intracellular infection of murine RAW264.7 macrophages. To accurately identify genes required for growth with a transposon library, it is important to sequence the input sample as well as the output sample used for each experiment. As confirmed earlier, every time a transposon library is passaged in LB medium, mutants that exhibit reduced fitness

will be lost from the library. Therefore, we sequenced two biological replicates of the library that had been grown in LB prior to infection of macrophages (Input 1 and Input 2 samples). Proliferation of the transposon library was assessed after a single passage through murine macrophages, as additional passages caused the selection of LPS mutants (S1 Text, S3 Fig). Accordingly, at 12 h post-infection (p.i.), intracellular bacteria from the two biological replicates were recovered (Macrophage 1 and Macrophage 2 samples) (Fig 1, S1 Text). Genomic DNA from the two input and the two output samples was purified and Illumina sequenced. S1 Table contains the number of demultiplexed reads, the number of reads with a transposon tag, and the number of uniquely mapped reads. The two input samples contained 660,000 and 928,000 unique insertion sites. Combining these results with the analysis of the previous input dataset (797,000 unique insertion sites), a total of 511,000 unique insertion sites were common to all three datasets, an average of one transposon integration per ten nucleotides.

The majority of genes (87%) that were designated "required" in the two new inputs were consistent with the analysis of the previous input sample (S4A Fig, S4 Table). Eleven of the genes considered ambiguous in the previous input sample were required in Input 1 and Input 2, including $cI^{BTP1}$, the repressor of the D23580-specific prophage BTP1 [28]. Prophage repressors are commonly found to be required genes for growth in TIS studies because inactivation leads to prophage de-repression and phage-mediated cell lysis. Among the five complete prophage regions in D23580 (BTP1, Gifsy-$2^{D23580}$, ST64B$^{D23580}$, Gifsy-$1^{D23580}$, and BTP5) [28], only the repressors of the two D23580-specific prophages, BTP1 and BTP5, were designated as required in our analyses.

The data were used to identify genes that contributed to fitness of *S*. Typhimurium D23580 during macrophage infection. Specifically, genes important for intracellular growth and survival in murine macrophages were identified by comparing the two macrophage output samples with the two input samples (Materials and Methods). Transposon insertions in 206 D23580 genes caused attenuation in the macrophage infection model (log$_2$ fold-change $<$-1, *P*-value $<$0.05) (S7 Table). Many of these genes correlated well with previous high-throughput studies of *S*. Typhimurium ST19 in different animal infection models (S5A and S5B Fig), including five well-characterized regulatory systems that control *Salmonella* virulence: the *phoPQ* two-component regulators; the *ssrAB* regulators of SPI-2 gene expression; *dam*, DNA adenine methylase; *hfq*; and *ompR*, an element of the two-component regulatory system *ompR-envZ*. Three D23580-specific genes, two in the pBT1 plasmid (*pBT1-0081* and *pBT1-0401*) and $cI^{pBT1}$, had already been identified as important for fitness after three passages in LB in our study.

Inactivation of six D23580 genes increased fitness in the macrophage infection model (log$_2$ fold-change $>$1, *P*-value $<$0.05): *nadD*, encoding a nicotinate-nucleotide adenylyltransferase; *STM1674* (*STMMW_16691*), encoding a transcriptional regulator; *barA*, encoding the sensor of the two-component regulatory system SirA/BarA that controls carbon metabolism via the CsrA/CsrB regulatory system; the pSLT-BT plasmid gene *repC;* and the LPS O-antigen biosynthetic genes *abe* (*rfbJ*) and *rmlA* (*rfbA*). Because the *abe* and *rmlA* mutants have short LPS which increases the invasiveness of *S*. Typhimurium without affecting intracellular replication [48], it is clear that our TIS strategy in macrophages not only identified mutants with increased fitness in terms of growth and survival inside macrophages but also selected for mutants that are more invasive in the infection model (S1 Text).

## Macrophage-specific *S*. Typhimurium D23580 genes

To identify genes important for fitness inside macrophages but not for growth in laboratory media, the 206 D23580 genes that showed attenuation in macrophages when disrupted by a

transposon insertion were cross-referenced with genes required for growth in the *in vitro* laboratory conditions tested in this study, LB and InSPI2 (S4B Fig, S4 Table). We identified 182 "macrophage-associated genes" and, within this group, 68 "macrophage-specific genes" that had reduced fitness during macrophage infection and did not impact upon growth *in vitro*. The macrophage-specific genes included known *Salmonella* virulence genes: *phoPQ*, SPI-2 genes, *dam*, *hfq*, and *ompR*. Most of the 68 genes were involved in functions related to transcription (10%), amino acid transport and metabolism (9%), and translation, ribosomal structure and biogenesis (7%).

Analysis of our intra-macrophage transcriptome of *S*. Typhimurium ST19 showed that genes that encoded key virulence factors were macrophage-up-regulated by >3-fold [49]. Our recent D23580 RNA-seq results [15] led us to investigate the function of two genes in the macrophage infection model, *STM2475* and *STM1630*. Only the *STM2475* deletion mutant exhibited decreased intracellular replication, suggesting a putative role in virulence of D23580 (S1 Text, S6 and S7 Figs).

We previously showed that many *S*. Typhimurium virulence genes were both up-regulated within macrophages and required for animal infection [50]. We built on this concept by finding the "macrophage-specific" genes identified by transposon mutagenesis that were also up-regulated within macrophages, using transcriptomic data from Canals and colleagues [15] (Fig 5A). The results showed that the 23 genes that were required for intra-macrophage proliferation and were significantly "macrophage-up-regulated" (fold-change >2, False Discovery Rate (FDR) <0.001) encoded: 14 SPI-2 proteins; two phosphate transport proteins (PtsB and PtsC); two enzymes involved in the arginine biosynthesis pathway (ArgB and ArgC); and the proteins Fis (DNA-binding protein), IolR (repressor of *myo*-inositol utilization), RluD (pseudouridine synthase), WzxE (translocation of the enterobacterial common antigen to the outer membrane), and OmpR. We searched for genes that had not been previously been reported to play a role in virulence in *S*. Typhimurium ST19 [51,52] and found only three: a SPI-2 gene (*sscB*) and *argBC*.

To determine if the requirement for the arginine biosynthetic pathway during intra-macrophage replication was a specific feature of the D23580 strain, mutants in the *argA* gene were constructed in D23580 and the ST19 strain 4/74. ArgA is the first enzyme for the biosynthesis of L-arginine from L-glutamate, a pathway that also includes the biosynthesis of L-ornithine [53]. Of the nine genes encoding products involved for the L-arginine biosynthesis, four were in the 68 macrophage-specific gene list: *argA*, *argCB*, *argE*. Furthermore, the encoded products are also involved in the L-ornithine biosynthetic sub-pathway. The individual Δ*argA::frt* mutants of both strains, D23580 and 4/74, showed reduced intra-macrophage replication (Fig 5B). The importance of ArgA for growth inside J774 macrophages has already reported for the *S*. Typhimurium ST19 isolate 14028 [54]. Our results indicate that the requirement for arginine genes inside murine macrophages is not a distinguishing feature of D23580 because the same genes were also required by the ST19 isolate 4/74. The decreased ability of the Δ*argA::frt* mutants to proliferate is consistent with previous studies, and suggests that arginine is a limiting factor for *S*. Typhimurium growth inside murine macrophages. The requirement for ArgA for optimal intra-macrophage replication of D23580 validates our TIS-based approach in this infection model.

Taken together, the 68 macrophage-specific gene list included many genes that encode *S*. Typhimurium ST19 virulence factors, and did not include any D23580-specific genes. We conclude that no novel virulence factors required for intra-macrophage replication of *S*. Typhimurium ST313 were identified in our experiments.

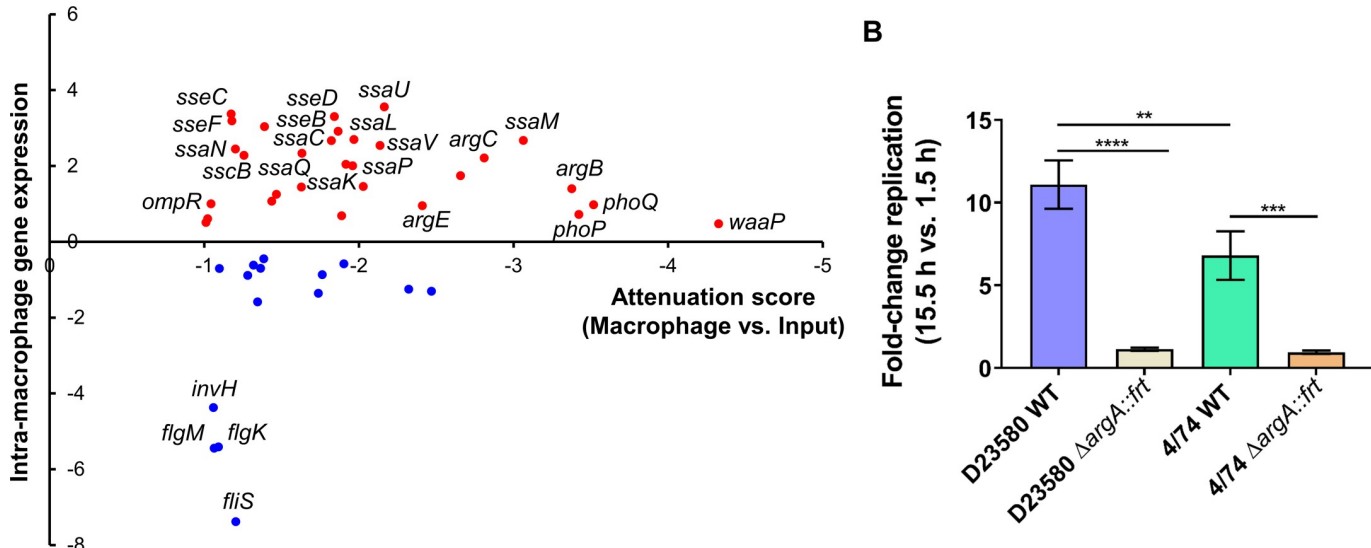

**Fig 5. Macrophage-specific genes of *S.* Typhimurium D23580 are required for virulence in animal infection models.** (A) Representation of the 45 genes, among the 68 macrophage-specific genes, in *S.* Typhimurium D23580 that showed an RNA-seq fold-change with FDR ≤0.05. The RNA-seq fold-change was calculated comparing transcriptomic data from D23580 recovered from the intra-macrophage environment (8 h post-infection) versus D23580 grown to ESP in LB (extracted from Canals and colleagues [15]). The attenuation score represents $\log_2$ fold-change of the TIS data obtained comparing the two output Macrophage samples versus the two Input samples. (B) Fold-change replication (15.5 h versus 1.5 h) in murine RAW264.7 macrophages. Average of three independent biological replicates each. Error bars show standard deviation. ****, *P*-value <0.0001; ***, *P*-value = 0.0006; **, *P*-value = 0.0043.

## Intergenic regions and sRNAs important for fitness inside murine macrophages

To identify *S.* Typhimurium D23580 intergenic regions and sRNAs that impact upon fitness inside macrophages but not in growth in the laboratory media LB and InSPI2, transposon insertions in short genomic regions were investigated (Materials and Methods). Transposon insertions in ten intergenic regions caused macrophage-specific attenuation (S4C Fig). Four of them were located in the plasmid regions. In pSLT-BT, the intergenic regions included: *spvB-spvA*, upstream of *spvB*, which encodes an ADP-ribosyltransferase that destabilizes actin polymerization of the host cells [55,56]; *int-dhfrI* encode an integrase and a trimethoprim resistance gene cassette in the Tn*21*-like element; and the downstream region of *repA* (called *repA_2* in D23580). In the pBT1 plasmid, the intergenic region was located between two genes encoding hypothetical proteins, *pBT1-0171* and *pBT1-0181*. Assuming disruption of promoter regions of the flanking genes, most of the chromosomal intergenic regions were expected to have an effect in fitness inside macrophages: upstream of *dksA*, upstream of a gene located in a SPI-6 associated region, SPI-2, upstream of *pssA* (phosphatidylserine biosynthesis), and upstream of *rpoB* (DNA-directed RNA polymerase subunit beta). An exception was seen upstream of *rmlB* (biosynthesis of O-antigen), where the effects on fitness were likely due to the polarity effects of the insertion of a strong promoter upstream of this gene. Overall, the phenotype of most of these intergenic transposon insertions was supported by previous studies that showed that the particular downstream genes were important for growth and intracellular survival inside macrophages.

Only transposon insertions in one sRNA caused attenuation in the intra-macrophage environment but not in the LB and InSPI2 *in vitro* growth conditions, namely AmgR (S4D Fig). This sRNA is an antisense RNA of the *mgtC* gene [57]. AmgR attenuates virulence mediated by decreasing MgtC protein levels [57]. The fact that disruptions in this sRNA attenuate

D23580 within macrophages should be interpreted with caution because transposon insertions disrupt both DNA strands and AmgR overlaps the *mgtC* gene, which is known to be critical for macrophage survival [58]. This finding may simply reflect the known role of *mgtC* in macrophage infection and not the involvement of AmgR in modulating MtgC protein levels.

## Limitations of this study

As for all global mutagenesis approaches, it is important to consider the limitations of our strategy. First, the relatively large size of the mutant pools generated a highly competitive environment, in which trans-complementation could occur. This phenomenon is characterized by compensating genetic defects in some mutants by the presence of the functional genes in other mutants. Second, because the Tn5 transposon carries an outward facing promoter that drives expression of the Km resistance gene, individual transposon insertions can cause polar effects due to the increased transcription of downstream genes [59,60]. Third, in the case of macrophage infection, although it would be ideal if individual macrophages were only infected by a single Tn5-carrying bacterium, the final multiplicity of infection (M.O.I.) was on average 42:1, meaning that combinations of mutants could have co-localized within the same intra-macrophage vacuole. The genes that contribute to intra-macrophage fitness that were identified here reflected selection for mutants with defects in the ability to replicate and survive inside macrophages, and also selection for mutants lacking certain SPI-1-associated factors such as InvH [61]. The genes required for optimal intra-macrophage fitness of *S.* Typhimurium sequence type ST313 showed substantial overlap with *S.* Typhimurium ST19 genes previously associated to virulence.

In principle, transposon orientation bias could lead to polarity effects of the insertion in the target gene or surroundings, resulting in some incorrect assignments of fitness effects to specific genes. Examples of transposon insertion bias have been discussed in previous TIS studies [62], but comprehensive analyses are scarce for this type of data [63]. To determine whether transposon orientation bias had impacted upon our data, a transposon orientation score was calculated for each gene, sRNA, rRNA, and intergenic region for all samples as discussed in detail in S1 Text (S8 Table). In summary, for those protein-coding genes for which sufficient data were available, the vast majority (>99%) showed no evidence of a strong influence of transposon orientation on mutant fitness.

## *S.* Typhimurium D23580 has many plasmid-encoded required genes

*S.* Typhimurium D23580 contains four plasmids: pSLT-BT (~117 kb), pBT1 (~84.5 kb), pBT2 (~2.5 kb), and pBT3 (~2 kb) [14,15]. The pSLT-BT and pBT1 plasmids have published annotations that were used to study gene requirements for growth and survival in this study. Two pSLT-BT plasmid genes were designated as required, *parA* and *parB* (Fig 6A, S2 Table), and the same genes were also found to be required for pBT1 plasmid maintenance (Fig 6B, S2 Table). The requirement of ParA and ParB for effective plasmid partitioning means that transposon insertions in both *parA* and *parB* caused plasmid loss in previous TraDIS studies [64]. The unexpected discovery of 34 more pBT1-encoded genes that were required, for stable maintenance of the plasmid or optimal fitness of *S.* Typhimurium D23580, included the *cysS*$^{pBT1}$ gene that has been reported previously [15].

The pBT1 plasmid was dispensable for *in vitro* growth of *S.* Typhimurium D23580 in LB (Fig 7A, S3 and S5 Tables) and required for optimal intra-macrophage replication (S1 Text, S6 Fig) [15]. The fact that our TIS analysis identified so many pBT1-encoded genes that were required suggests the involvement of multiple genes in the maintenance and replication of the plasmid [64,65]. For example, the *repA*$^{pBT1}$ gene, involved in plasmid replication [66], was

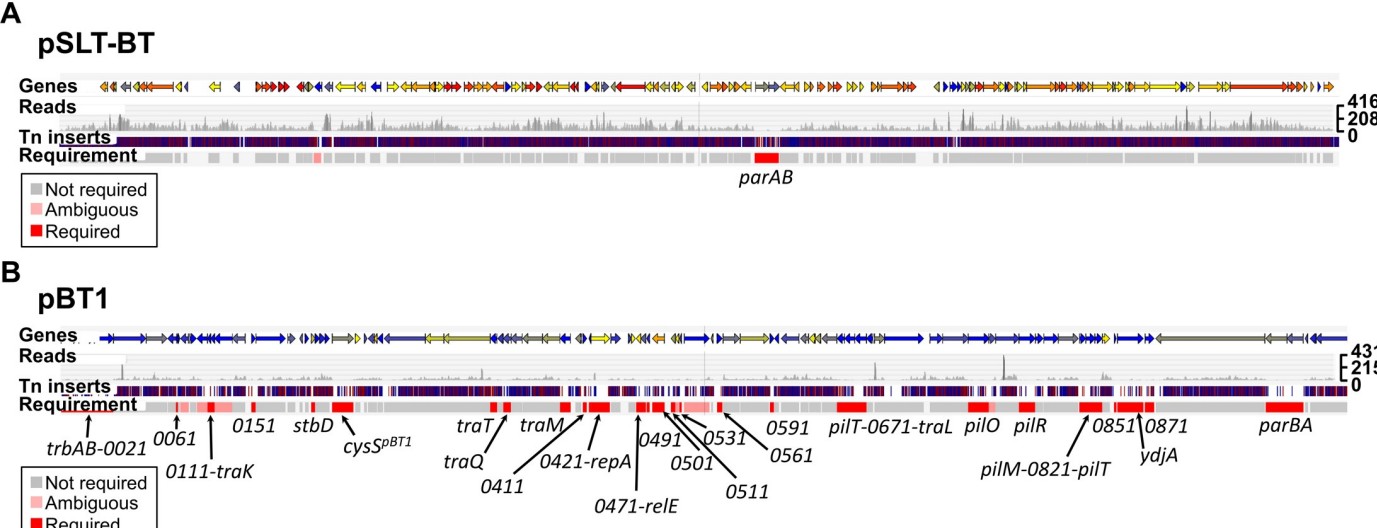

**Fig 6. Identification of required genes encoded on *S*. Typhimurium D23580 pSLT-BT and pBT1 plasmids.** (A) pSLT-BT plasmid; and (B) pBT1 plasmid. Figures were obtained using the Dalliance genome viewer (https://hactar.shef.ac.uk/D23580). Coloured arrows at the top represent genes (colour is based on GC content, blue = low, yellow = intermediate, red = high). Each sample is represented by three tracks, in this case the Input is the only sample shown. The first track shows raw data for the Illumina sequencing reads. The second track contains blue and red lines that correspond to transposon insertion sites; red = + orientation of the transposon, same as genes encoded on the plus strand, blue = opposite orientation. The third track highlights in red those genes that were considered required for growth in that condition based on an insertion index (Materials and Methods). The names of required genes are indicated at the bottom. The scale on the right represents sequence read coverage.

required for pBT1 in D23580. The high proportion of pBT1 genes (38%) designated required could reflect the presence of particular features in this plasmid, such as toxin/antitoxin systems and/or DNA-binding proteins. We noted that the percentage of AT content of the pBT1 plasmid is particularly high, 56.7%, compared to the *S*. Typhimurium D23580 chromosome and the pSLT-BT plasmid, which are 47.8% AT and 46.5% AT, respectively. The fact that pBT1 is so AT-rich parallels the AT content of the pSf-R27 plasmid of *Shigella flexneri* 2a strain 2457T which is 55%. An H-NS paralogue, Sfh, is encoded by pSf-R27 and is responsible for a stealth function that allows the plasmid to be transmitted to new bacterial hosts with minimal effects on fitness [67,68]. Consequently, the introduction of a modified version of the plasmid that lacked *sfh* (pSf-R27Δ*sfh*) into *S*. Typhimurium ST19 SL1344 significantly decreased fitness due to interference with the H-NS regulatory network, whereas the wild-type pSf-R27 plasmid itself did not impact upon fitness when introduced into the same strain [69]. This parallel between the pBT1 and pSf-R27 plasmids raises the possibility that pBT1 encodes an H-NS-like protein that has a global impact upon fitness of D23580, but could not be found by sequence identity alone.

To investigate pBT1-specific features that could explain the high number of genes with lower amount of transposon insertions than the average amount in the chromosomal and the pSLT-BT plasmid, genes with annotated known functions were studied. The pBT1 plasmid carries a set of conjugative genes and was successfully introduced into 4/74 by conjugation demonstrating the functionality of those genes. At least 11, out of the 36 required genes in pBT1, encoded putative conjugal proteins that are critical for plasmid transfer (*trbAB*, *traK*, *traT*, *traQ*, *traM*, *pilT*, *traL*, *pilO*, *pilR*, *pilM*, *pilT*) [70]. The transposon-induced overexpression of the *traAB* genes in IncI plasmids has been reported to cause a growth defect [71]. Additionally, the plasmid contains at least one annotated toxin/antitoxin system: *stbD* (*pBT1-0201*)/ *stbE* (*pBT1-0211*). The gene encoding the antitoxin StbD was required, while the gene encoding the

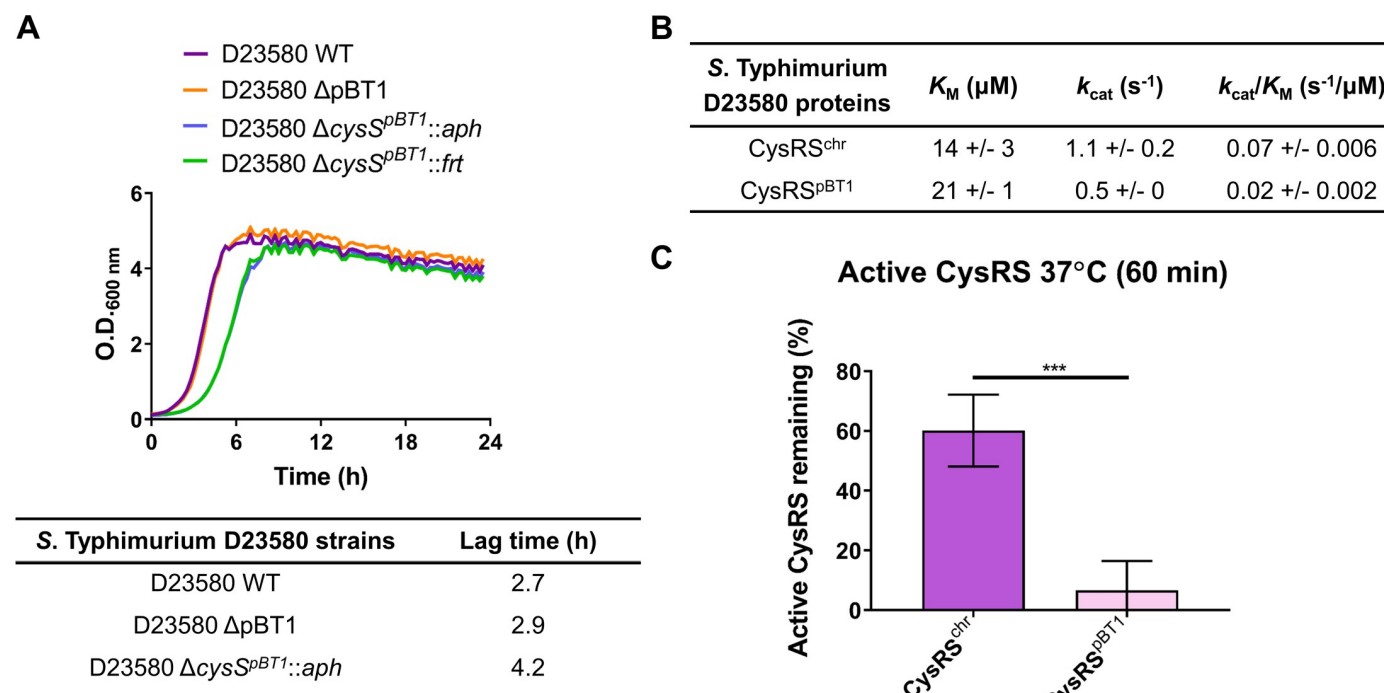

**Fig 7. The pBT1 plasmid-encoded CysRS^PBT1^ is less efficient and stable than the chromosomal-encoded CysRS^chr^.** (A) Growth curves in LB medium of the D23580 WT strain, the pBT1-cured D23580 strain, and the Km resistant (*aph*) and deletion versions (*frt*) of the *cysS^pBT1^* mutant. The table highlights differences in the lag time between strains. (B) The catalytic efficiency ($k_{cat}/K_M$) of CysRS^PBT1^ is 3-fold lower than for CysRS^chr^. Recombinant CysRS^PBT1^ and CysRS^chr^ were purified by overexpression in *E. coli*. ATP-PPi exchange was used to determine the steady-state kinetic parameters for activation of cysteine. (C) CysRS^PBT1^ is 10-fold less stable than CysRS^chr^. This comparison was performed by incubating the enzymes at 37˚C for 60 minutes.

toxin StbE was dispensable (S2 Table). Two other genes, *relE* (*pBT1-0481*) and *relB* (*pBT1-0521*), are annotated as a toxin and an antitoxin, respectively, and are separated by three genes. Our data showed that the gene encoding the toxin RelE was required, while the gene encoding the antitoxin RelB gave an ambiguous result. The precise role of pBT1-encoded toxin/antitoxin systems upon plasmid biology and the fitness of D23580 merits further investigation.

One of the limitations of TIS approaches is that polarity effects can be caused by the introduction of a strong transposon-encoded promoter in a specific genetic context. Such polarity effects could explain the low density of transposons observed in some regions of the pBT1 plasmid, compared with the pSLT-BT plasmid and other plasmids studied in previous TIS studies. Most of the pBT1-encoded genes are hypothetical, and lack a known function. It remains to be determined if the high expression of certain plasmid regions could be toxic to bacterial cells.

The finding that curing of pBT1 does not impact upon fitness shows that the plasmid is not in itself essential. Previously, the observation of fitness defects associated with mutations in specific plasmid-encoded genes reflected the essentiality of the entire plasmid [72,73], which is extremely uncommon [74]. Because the fitness defect of the D23580 Δ*cysS^pBT1^* mutant could reflect an uncharacterized aspect of pBT1 biology, it was investigated in more detail.

## The pBT1-encoded CysRS^PBT1^ is less efficient and stable than the chromosomal-encoded CysRS^chr^

Cysteinyl-tRNA synthetase (CysRS) is an essential protein for bacterial translation. Recently, we showed that *S.* Typhimurium D23580 expresses high levels of *cysS^pBT1^* and that the *cysS^chr^*

is expressed at very low levels [15]. To investigate why an organism would express a plasmid-encoded gene instead of a chromosomal copy, recombinant CysRS$^{chr}$ and CysRS$^{pBT1}$ proteins were purified by overexpressing the relevant D23580 genes in *E. coli* and studied enzymatically. The steady-state kinetic parameters for activation of cysteine by both CysRS$^{chr}$ and CysRS$^{pBT1}$ were determined using ATP-PP$_i$ exchange. CysRS$^{pBT1}$ had a 3-fold lower catalytic efficiency ($k_{cat}/K_M$) than CysRS$^{chr}$ (Fig 7B). Additionally, the stability of the enzymes was determined. CysRS$^{pBT1}$ was ten times less stable than CysRS$^{chr}$ after incubation at 37˚C for 60 min (Fig 7C). The fact that CysRS$^{pBT1}$ was both less efficient at cysteine activation and more unstable over time raises the possibility that *S.* Typhimurium D23580 could be using CysRS$^{pBT1}$ as a trigger to shut down translation during stressful conditions. A less efficient CysRS within the bacterial cell would lead to the accumulation of uncharged tRNAs which could induce the stringent response to help the cell cope with stress [75,76]. We speculate that the preference for expressing the plasmid-encoded *cysS$^{pBT1}$* could give *S.* Typhimurium D23580 the ability to respond to stress conditions at the level of translation. Clearly, further research is required to determine whether the distinct biochemical properties of CysRS$^{pBT1}$ influence the pathogenesis of African *S.* Typhimurium ST313.

To investigate how widespread plasmid-encoded aminoacyl-tRNA synthetases are in biology, a database containing 13,661 bacterial plasmids was generated (Materials and Methods). A total of 79 plasmid gene products were found to contain tRNA synthetase-related functional annotations (Fig 8). Closer inspection of those CDS (coding sequences) revealed two classes: complete aminoacyl-tRNA synthetase CDS (30); and CDS encoding aminoacyl-tRNA synthetase fragments and/or related functions (49) (S9 Table). The presence of aminoacyl-tRNA synthetase paralogues and paralogous fragments have been previously reported in eukaryotic and prokaryotic genomes [77]. Our analysis suggests that alternate plasmid-encoded aminoacyl-tRNA synthetases exist, and the molecular function of plasmid-encoded translation-related genes warrants further study.

## Perspective

The significant impact of iNTS disease as a major public health problem in sub-Saharan Africa has led to *S.* Typhimurium ST313 becoming an active focus of research. To understand how African *S.* Typhimurium ST313 causes disease, it was important to determine whether this pathovariant carries novel virulence genes that had not been previously described in *S.* Typhimurium ST19. Overall, this work found relatively few differences in genetic requirements for growth between an African *S.* Typhimurium ST313 strain and gastroenteritis-associated *S.* Typhimurium ST19 strains that have been studied previously. Our transposon insertion sequencing approach during infection of the murine RAW264.7 macrophage infection model suggests that *S.* Typhimurium ST313 D23580 does not encode novel virulence factors. However, only two infection-relevant *in vitro* conditions and intra-macrophage replication were studied. For the future, it will be important to explore the genetic requirements of *S.* Typhimurium ST313 in more environmental conditions.

Here, we present an online resource that allows the candidate genes that impact upon fitness of *S.* Typhimurium ST313 strain D23580 to be visualized, both in particular *in vitro* growth conditions and during intra-macrophage replication: https://hactar.shef.ac.uk/D23580.

In terms of plasmid biology, we conclude that, although pBT1 is not essential for growth of *S.* Typhimurium ST313 strain D23580, the plasmid contains a high proportion of genes that impact upon bacterial fitness. Our findings raise the possibility that plasmid-encoded aminoacyl-tRNA synthetases play a hitherto unrecognised role in bacterial physiology.

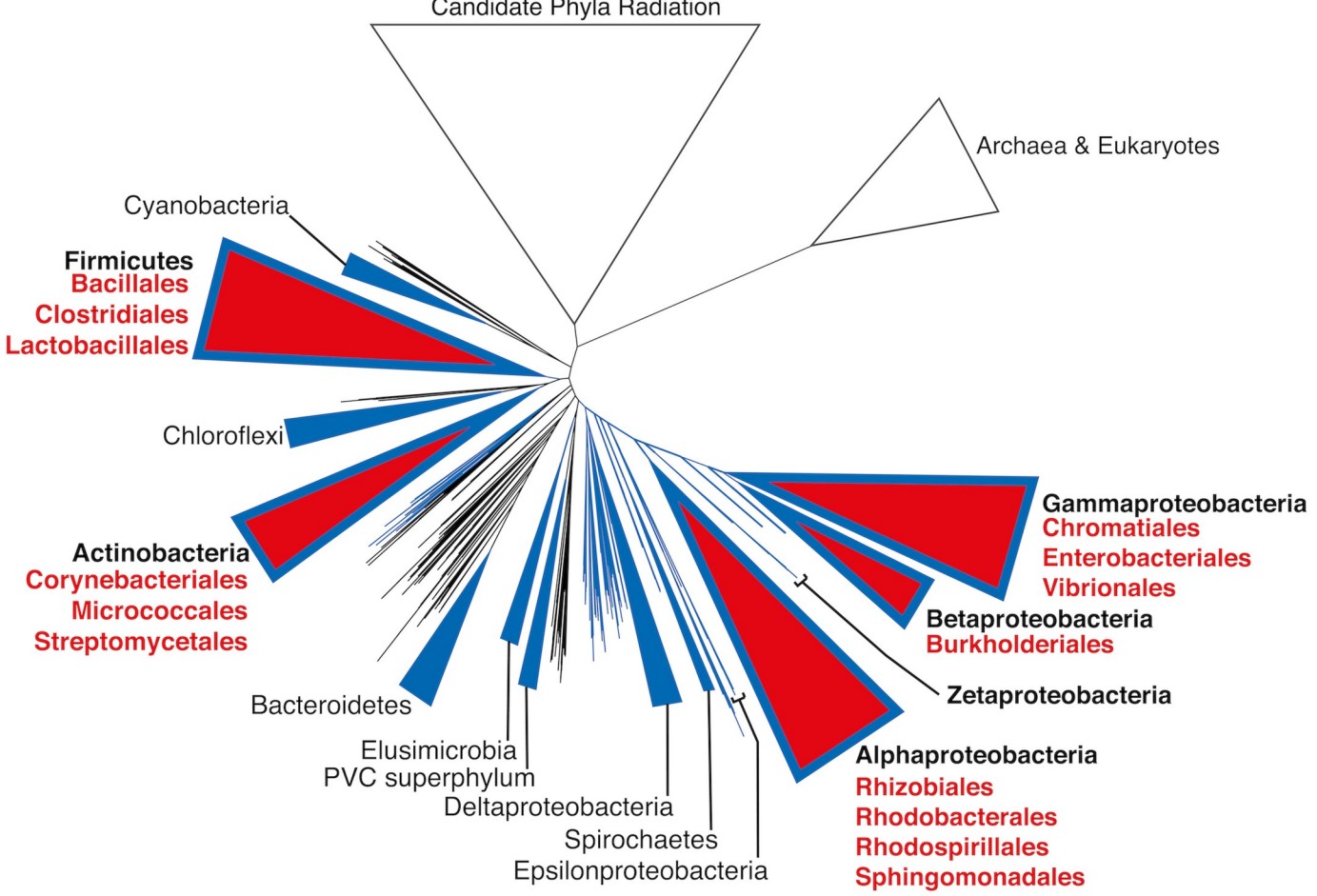

**Fig 8. Putative plasmid-encoded aminoacyl-tRNA synthetase genes are associated with diverse Gram-negative and Gram-positive bacteria.** The phylogenetic distribution of putative tRNA synthetase genes detected in our plasmid database is displayed in the context of the ribosomal protein tree of life constructed by Hug and colleagues [78]. Large phyla or classes are collapsed into triangles. Taxa represented by our database of 13,661 plasmid sequences are displayed in blue, and the phylogenetic distribution of the 79 putative plasmid-encoded aminoacyl-tRNA synthetase genes is indicated in red (S9 Table).

## Materials and methods

### Bacterial strains and growth conditions

LB was obtained by mixing 10 g/L tryptone (Difco), 5 g/L yeast extract (Difco), and 5 g/L NaCl (Sigma). The InSPI2 medium was based on PCN (pH 5.8, 0.4 mM Pi), which was prepared as previously described [79]. When required, the antibiotic Km was added to a final concentration of 50 μg/mL, and tetracycline (Tc) to 20 μg/mL.

Bacterial strains used for this study are shown in S10 Table. Permission to work with *S.* Typhimurium strain D23580 from Malawi [14] was approved by the Malawian College of Medicine (COMREC ethics no. P.08/4/1614).

## Construction of a transposon library in *S.* Typhimurium D23580

A library of transposon insertion mutants was constructed in *S.* Typhimurium D23580 as previously described with some modifications [21]. Briefly, D23580 was grown in rich medium to logarithmic phase and competent cells were prepared. Transposome mixtures were prepared by mixing the EZ-Tn5 <KAN-2> transposon from Epicentre Biotechnologies, EZ-Tn5 transposase and TypeOne restriction inhibitor, and transposomes were transformed into D23580 competent cells. A total of eight electroporations derived from two transposome mixtures were performed and cells were recovered by addition of SOC medium and incubation at 37°C for 1 h. Bacterial mixtures were plated onto LB agar Km at a concentration of 50 μg/mL and incubated at 37°C overnight. The transposon mutants were collected from the plates by adding LB and were joined to grow them together in LB Km 50 μg/mL at 37°C overnight.

## Passages of the *S.* Typhimurium D23580 transposon library in LB and InSPI2

The D23580 transposon library was grown in LB Km 50 μg/mL at 37°C 220 rpm for 16 h, and genomic DNA was purified from a fraction of the bacterial culture (Input). Another fraction of the bacterial culture was washed twice with PBS and resuspended in LB or InSPI2 media. A dilution 1:100 (~2.5 x $10^8$ cells) was inoculated into 25 mL of LB or InSPI2 media (without antibiotic), respectively, and cultures were incubated at 37°C, 220 rpm for 24 h (passage 1). Two more passages were performed in an InSPI2 media. For LB passages, 250 μL were transferred in each individual passage, after two washes with PBS (~1.4 x $10^9$ cells). For InSPI2 passages, 640 μL were transferred into the next passage (~1.7 x $10^8$ cells). Genomic DNA was purified from the third passage in LB (output LB) and InSPI2 (output InSPI2). Genomic DNA purifications were performed using the DNeasy Blood & Tissue Kit (Qiagen) following manufacturer's indications for extractions from Gram-negative bacteria.

## Library preparation and Illumina sequencing for the input and the LB and InSPI2 output samples

Genomic DNAs were fragmented to 300 bp with a S220/E220 focused-ultrasonicator (Covaris). Samples were prepared using the NEBNext DNA Library Prep Master Mix Set for Illumina for use with End User Supplied Primers and Adapters (New England Biolabs) following manufacturer's instructions. The DNA fragments were end-repaired and an "A" base was added to the 3'ends prior to ligation of the Illumina adapters (*PE Adapters* from Illumina). In order to amplify the transposon-flanking regions, transposon-specific forward oligonucleotides were designed such that the first 10 bases of each read would be the transposon sequence: PE PCR Tn-12, for the input sample (Input); PE PCR Tn-1 and PE PCR Tn-7, for the output sample in LB; and PE PCR Tn-10 and PE PCR Tn-4, for the output sample in InSPI2 (S11 Table). These oligonucleotides were used for PCR amplification with the Illumina reverse primer *PE PCR Primer 2.0*. The primers included the adapters and the sequences necessary for attachment to the Illumina flow cell. Furthermore, a specific 6-base barcode included in the forward primer was incorporated into each of the samples in order to pool them together in a single lane for sequencing. Oligonucleotides ordered were HPLC-purified with a phosphorothioate bond at the 3'end from Eurofins Genomics. The three libraries of amplified products were pooled in equimolar amounts and size-selected to 200–500 bp. After QC assessment, the pool was paired-end sequenced, using the Illumina sequencing primers, in one lane on a HiSeq 2500 at 2x125 bp. 15% of library of the bacteriophage ΦX174 genome, provided by

Illumina as a control, was added to the lane to overcome the low complexity of the bases after the barcode in Read 1.

## Sequence analysis of the *S*. Typhimurium D23580 transposon library

Cutadapt version 1.8.1 was used to demultiplex the sequence reads based on the 6-base barcode [80]. Transposon sequences at the beginning of the reads were removed using the same program. BWA-MEM [81] was used to map the reads against the *S*. Typhimurium D23580 genome sequence (accession: PRJEB28511). Reads with a mapping quality <10 were discarded, as were alignments which did not match at the 5'end of the read (immediately adjacent to the transposon) (S1 Table). The exact position of the transposon insertion sites and the frequency of transposons for every annotated gene were determined.

An insertion index was calculated for each gene as explained in [20], using the Bio-Tradis toolkit [82]. Genes with insertion index values <0.05 (cut-off determined using Bio-Tradis) were considered as "required" for growth in the Lennox rich medium. Genes with an insertion index between 0.05 and 0.075 were considered "ambiguous", and genes with an insertion index >0.075 were considered "not required". S2 Table shows the number of reads, transposon insertion sites, insertion index, and essentiality call per gene.

For the transposon orientation analysis, the total number of unique insertion sites for each gene and intergenic region were counted once (*all*) in the input and output samples, regardless of the number of reads associated with each insertion position (S8 Table). The number of insertions in the same orientation as the plus DNA strand (*top*) and the minus DNA strand (*rev*) were calculated irrespective of gene orientation. Additionally, the number of insertions in the same orientation as the gene direction were represented as values that indicated either *sense* (same orientation as gene direction) or *anti* (opposite orientation as gene direction). For intergenic regions, the *sense* and *anti* values were arbitrarily attributed to the same orientation as the plus and the minus DNA strands, respectively. A transposon orientation score was calculated for both analyses: (*top—rev*)/*all* and (*sense—anti*)/*all*, respectively; using a modified approach from a recent study [63]. The values in the (*sense—anti*)/*all* column were used as the transposon orientation scores for the analysis. *P*-values were calculated using a Chi-squared test with a null hypothesis of equal probability of insertion in either orientation, and these were corrected for multiple testing using FDR (*adjP*).

## Infection of RAW264.7 macrophages with *S*. Typhimurium D23580

The D23580 transposon library was grown in LB Km 50 μg/mL at 37°C 220 rpm for 16 h, and genomic DNA was purified from two different biological replicates as input samples. Murine RAW264.7 macrophages (ATCC TIB-71) were grown in DMEM high glucose (Thermo Fisher Scientific) supplemented with 10% heat-inactivated fetal bovine serum (Thermo Fisher Scientific), 1X MEM non-essential amino acids (Thermo Fisher Scientific) and 2 mM L-glutamine (Thermo Fisher Scientific), at 37°C in a 5% $CO_2$ atmosphere. $10^6$ macrophage cells were seeded on each well of 6-well plates (Sarstedt) 24 h before infection. Bacteria were opsonized with 10% BALB/c mouse serum (Charles River) in 10 volumes of DMEM for 30 min on ice. Macrophages were infected with *Salmonella* at an M.O.I. of approximately 10:1, and infections were synchronized by centrifugation (5 min at 1,000 rpm). After 30 min of infection, macrophages were washed with DPBS (Thermo Fisher Scientific), and DMEM with supplements and gentamicin 100 μg/mL was added to kill extracellular bacteria. After 1 h, macrophages were washed with DPBS, and fresh DMEM with supplements and gentamicin 10 μg/mL was provided for the rest of the incubation time at 37°C with 5% $CO_2$. In some wells, 1% Triton X-100 was added to recover intracellular bacteria and plate dilutions to obtain bacterial counts

for the 1.5 h time point. For the rest of the wells, after 12 h from the initial infection, macrophages were washed with DPBS and intracellular bacteria were collected using 1% Triton X-100. Some wells were used for obtaining bacterial counts and calculate the fold-change replication of the intracellular bacteria (12 h versus 1.5 h) (S3B Fig). For the output samples of the D23580 transposon library, 12 wells (of 6-well plates) were pooled for each replicate. The samples of 1% Triton X-100 containing the intracellular bacteria were centrifugated and the supernatants were discarded. Pellets were resuspended in LB and transferred into a flask to grow bacteria for 10 h 220 rpm in LB supplemented with Km. Genomic DNA was extracted from those cultures and prepared for Illumina sequencing, together with the input samples.

For macrophage infections with the individual *S.* Typhimurium strains D23580 WT, D23580 Δ*argA*::*frt*, 4/74 WT, and 4/74 Δ*argA*::*frt*, intracellular bacteria were recovered after 1.5 h and 15.5 h. Bacterial counts at the two time points were used to calculate the fold-change replication in the intra-macrophage environment (Fig 5B).

## Library preparation and Illumina sequencing for the input and output samples of the RAW264.7 macrophage experiment

Genomic DNA samples were prepared for Illumina sequencing following the previously described protocol. The transposon-specific forward oligonucleotides used for amplifying the sequence adjacent to the transposon were: PE PCR Tn-12, for the input biological replicate 1 sample (Input 1); PE PCR Tn-7, for the input biological replicate 2 sample (Input 2); PE PCR Tn-1, for the output biological replicate 1 sample (Macrophage 1); and PE PCR Tn-5, for the output biological replicate 2 sample (Macrophage 2) (S11 Table). In this case, oligonucleotides were ordered HPLC-purified with a phosphorothioate bond at the 3'end from Integrated DNA Technologies (IDT).

The four libraries of amplified products were pooled in equimolar amounts with two other libraries and size-selected and QC-assessed. The pool was paired-end sequenced, using the Illumina sequencing primers, in one lane on a HiSeq 4000 at 2x150 bp. In this case, 50% of a library of the bacteriophage ΦX174 genome was added to the lane to overcome the low complexity of the bases after the barcode in Read 1.

## Sequence analysis of the *S.* Typhimurium D23580 transposon library for the input and output samples of the RAW264.7 macrophage experiment

Analysis was performed using DESeq2 and following the same strategy described in [83]. Results are shown in $\log_2$ fold-change. A cutoff of 2-fold-change and *P*-value ≤0.05 was applied (S7 Table).

## Construction of mutants in *S.* Typhimurium D23580 and 4/74 by λ Red recombineering

Mutants were constructed using the λ Red recombination system [84]. Using oligonucleotides Fw-argA-P1 and Rv-argA-P2, a Km resistance cassette was PCR-amplified from plasmid pKD4. The PCR product was transformed by electroporation into D23580 and 4/74 containing the pSIM5-*tet* plasmid to replace the *argA* gene. Recombinants were selected on LB agar plates supplemented with Km. The 4/74 Δ*argA*::*aph* construction was transduced into WT 4/74 using the generalized transducing bacteriophage P22 HT105/1 *int-201* as previously described [28]. The antibiotic resistance cassettes from both, the 4/74 Δ*argA*::*aph* and D23580 Δ*argA*::*aph*, were removed by the use of the pCP20-TcR plasmid [85]. Strains and plasmids are included in S10 Table and oligonucleotides in S11 Table.

## CysRS cloning and purification

Genomic DNA from *S.* Typhimurium D23580 was used to PCR amplify both chromosomal and plasmid *cysS* which were then cloned into pET28a(+). Chromosomal and plasmid CysRS were expressed in *E. coli* BL21(DE3) with 1 mmol IPTG induction for 4 h. Cells were harvested, lysed by sonication and purified using a TALON metal affinity resin. CysRS was eluted with 250 mM imidazole and fractions containing protein were concentrated and dialyzed overnight in 50 mM Tris pH 7.5, 100 mM KCl, 5 mM $MgCl_2$, 3 mM 2-mercaptoethanol, 5% glycerol, and then dialyzed 4 h in similar buffer with 50% glycerol for storage. Proteins were stored at -20°C. Oligonucleotides used for cloning and expression are included in S11 Table.

## CysRS pyrophosphate exchange–steady state kinetics

To determine the $K_M$ for Cys, pyrophosphate exchange was completed in a reaction containing 100 mM HEPES pH 7.5, 30 mM KCl, 10 mM $MgCl_2$ 1 mM NaF, 25 nM CysRS, 50 μM-2mM Cys, 2 mM ATP, 2 mM $^{32}$P-PPi. The reaction without CysRS was incubated at 37°C for 5 min at which point the enzyme was added. Then aliquots were taken at 1–4 min by combining the reaction with quench solution (1% activated charcoal, 5.6% $HClO_4$, 1.25 M PPi). On a vacuum filter with 3 mm filter discs, filter discs were pre-rinsed with water, charcoal reaction added, washed 3x $H_2O$ and 1x 95% EtOH. Radiation was quantified using liquid scintillation counting. Michaelis-Menton equation was used to determine kinetic parameters.

## CysRS thermal stability

To determine the stability of protein, chromosomal and pBT1 CysRS were incubated at 37°C for 0 and 60 min and active site titration was used to measure the activity of the protein. Active site titration was completed in 30 mM KCl, 10 mM $MgCl_2$, 80 μM $^{35}$S-Cys, 2 mM ATP, pyrophosphatase and CysRS. At 0 min, 5 μL of CysRS were added to the reaction mixture and placed at 37°C for 10 min. The reaction was quenched by placing tubes on ice. After 60 min incubation of purified protein at 37°C, 5 μL were combined with the reaction mixture as above. All reactions were placed on a vacuum filtration unit on a Protran BA85 nitrocellulose membrane, washed three times with 1 mL 15 mM KCl and 5 mM $MgCl_2$ and dried. Then 4 mL of liquid scintillation cocktail were added and radiation was quantified using liquid scintillation counting.

## Growth curves of the D23580 bacterial strains

To determine the growth rate of the D23580 WT, D23580 ΔpBT1, D23580 Δ*cysS$^{pBT1}$*::*aph* and D23580 Δ*cysS$^{pBT1}$*::*frt* strains in LB, a Growth Profiler 960 was used (EnzyScreen). Bacterial cells grown for 16 h in LB, 37°C 220 rpm, were diluted in 250 μl of LB to $OD_{600}$ 0.01 and incubated in the Growth Profiler for 24 h at 37°C, shaking at 224 rpm. The $OD_{600}$ values were measured every 15 min.

## Statistical analyses

Graphpad Prism 8.0.1 was used for statistical analyses (GraphPad Software Inc., La Jolla, CA, USA). One-way ANOVA and Tukey's multiple comparison test were used for comparative analyses.

## Analysis of conservation of aminoacyl-tRNA synthetases in bacterial plasmids

A plasmid database was generated from the plasmid sub-section of the NCBI Reference Sequence Database (RefSeq) v90 [86] that included 13,924 complete plasmid sequences. The associated taxonomical information was downloaded and non-bacterial sequences were removed to leave 13,661 plasmids. The GenPept files were processed to extract the associated coding sequence annotations.

## Supporting information

**S1 Text. Supporting results and methods.**
(PDF)

**S1 Fig. Identification of genes that are required in *S.* Typhimurium D23580 but not in other *Salmonella* pathovariants.** (A) Comparative analysis of *S.* Typhimurium D23580 required genes with previously identified required genes in TIS studies in *S.* Typhimurium [21,25], and *S.* Typhi [20,21]. SL3261 was derived from SL1344; and WT174 was derived from Ty2. For the previously published studies, only genes that shared an ortholog in D23580 were included for the analysis. Individual Venn diagram analyses including a comparison with only *S.* Typhimurium (B) and *S.* Typhi (C) strains were also generated.
(TIF)

**S2 Fig. *S.* Typhimurium D23580 genes with reported H-NS binding sites in *S.* Typhimurium SL1344.** Individual growth curves, in LB medium, of the Km resistant versions (*aph*, $n = 8$) and the deletion versions (*frt*, $n = 7$) of (A) a SPI-1 mutant (*hilC*), a SPI-2 mutant (*ssrAB*), and (B) LPS mutants (*waaL* and *waaG*).
(TIF)

**S3 Fig. *S.* Typhimurium D23580 transposon library passaged three times in murine RAW264.7 macrophages.** (A) M.O.I. (number of bacterial cells used to infect one macrophage) of the D23580 transposon library infecting murine RAW264.7 macrophages for 8 h, used in the first ($n = 1$), second ($n = 3$) and third infections ($n = 3$). Fold-change replication of the intra-macrophage bacteria (8 h versus 1.5 h) of the D23580 transposon library seen after each passage. Error bars show standard deviation ($n = 3$). (B) M.O.I. and fold-change replication of the intra-macrophage bacteria of the D23580 transposon library at 12 h p.i. (C) Fold-change replication of the D23580 WT and 4/74 WT strains inside murine RAW264.7 macrophages. M.O.I.s calculated for one of the three biological replicates are indicated at the top of each bar ($n = 3$). (D) The percentage of rough mutants increased after passages of *S.* Typhimurium D23580 WT in macrophages and LB (first and second infections, $n = 1$; third infection, $n = 3$; after third passage, $n = 3$); (E) and in *S.* Typhimurium 4/74 WT (first and second infections, $n = 1$; third infection, $n = 2$; after third passage, $n = 3$).
(TIF)

**S4 Fig. 206 *S.* Typhimurium D23580 genes cause fitness alteration in macrophages.** (A) 10% of the *S.* Typhimurium D23580 genes are required for growth in all three LB input mutant pools. (B) Identification of *S.* Typhimurium D23580 "macrophage-specific" and "macrophage-associated" genes. The Venn diagram compares the 206 *S.* Typhimurium D23580 genes that showed attenuation in RAW264.7 macrophages when disrupted by a transposon insertion with required genes in the three inputs (Inputs), and the LB and InSPI2 outputs. (C) Venn diagrams including only intergenic regions, and (D) sRNAs.
(TIF)

**S5 Fig. Macrophage-attenuated genes of D23580 required for virulence of *S.* Typhimurium in other infection models.** (A) 63% of the D23580 macrophage-attenuated genes are important for virulence of *S.* Typhimurium 4/74 in food-related animal infection models [51]. Only orthologous chromosomal and pSLT plasmid genes were included for the analysis. (B) D23580 macrophage-attenuated genes compared to *S.* Typhimurium 14028 genes associated to virulence in BALB/c mice [52]. Only orthologous chromosomal genes were included for the analysis.
(TIF)

**S6 Fig. The *S.* Typhimurium D23580 Δ*STM2475* mutant and the pBT1-cured strain exhibited decreased proliferation in the intra-macrophage environment.** (A) Intra-macrophage proliferation assays of the D23580 WT, D23580 Δ*STM2475*::*frt*, D23580 *STM2475*$^{4/74SNP}$, D23580 ΔpBT1; and 4/74 WT, 4/74 Δ*STM2475*::*frt*, 4/74 *STM2475*$^{D23580SNP}$, 4/74 Δ*ssrAB*::*frt*. Bars represent average of three independent biological replicates and standard deviation. Significant differences indicate *P*-value: ***, 0.0002; **, 0.0011; *, 0.0116; ns, not significant. (B) Alignment of the *STM2475* promoter region in four *S.* Typhimurium strains. (C) Conservation of the nucleotide indel in *S.* Typhimurium ST313 strains.
(TIF)

**S7 Fig. The *STM1630* gene is inactivated in *S.* Typhimurium D23580.** (A) Transposon insertion profile of the *STM1630* region from our D23580 Dalliance genome browser. (B) There were not significant differences in growth in LB between D23580 WT and the Δ*STM1630* mutants, with or without the Km resistance cassette. (C) Absolute expression levels of *STM1630* in *S.* Typhimurium D23580 and 4/74 (extracted from Canals and colleagues [15]). Values represent TPM, TPM ≤10 means no expression. (D) Disruption of *STM1630* −10 box in the promoter region: two SNP-difference between 4/74 and D23580. (E) The D23580 isoform is conserved in all ST313 genomes analyzed, including lineage 1 and 2 and UK-ST313 strains described in Ashton and colleagues [87]. BLASTn was used to identify the genotype of the *STM1630* transcriptional start site −10 region in all genomes and the results were visualized in the context of the phylogenetic tree from Ashton and colleagues [87]. (F) Intra-macrophage proliferation assays of the D23580 and 4/74 WT strains, the Δ*STM1630*::*frt* mutants for D23580 and 4/74, and the D23580 *STM1630*$^{4/74SNP}$ mutant. Bars represent average of three independent biological replicates and standard deviation. Significant differences indicate *P*-value: ****, <0.0001; ***, <0.001; ns, not significant.
(TIF)

**S1 Table. Number of sequenced reads for each sample at every step.** Only R1 reads are included. The percentages were calculated relative to the previous step with the exception of deduplication, which was calculated relative to the number of mapped reads.
(XLSX)

**S2 Table. Number of reads, transposon insertion sites, insertion index, and essentiality call per gene.** Samples included: Input (for LB and InSPI2), LB (output), InSPI2 (output), Input 1 (for Macrophage), Input 2 (for Macrophage), Macrophage 1 (output), and Macrophage 2 (output).
(XLSX)

**S3 Table. Raw data for figures.**
(XLSX)

**S4 Table. Raw data for supporting figures.**
(XLSX)

**S5 Table. Lag time, maximum OD$_{600}$ and maximal growth rate from growth curves for different *S*. Typhimurium D23580 mutants and the WT strain.**
(PDF)

**S6 Table. Number of reads, transposon insertion sites, insertion index, and essentiality call per intergenic regions $\geq$100 bp and sRNAs.**
(XLSX)

**S7 Table. Analysis of the TIS macrophage data.** Read counts for the two inputs and two outputs (Macrophage), and log$_2$ fold-changes and adjusted *P*-values for the comparative analysis of each coding gene, noncoding sRNA, and intergenic regions in *S*. Typhimurium D23580.
(XLSX)

**S8 Table. Transposon orientation analysis.**
(XLSX)

**S9 Table. 79 aminoacyl-tRNA synthetases found in the custom bacterial plasmid database.**
(XLSX)

**S10 Table. Bacterial strains used in this study.**
(PDF)

**S11 Table. Oligonucleotides used in this study.**
(PDF)

## Acknowledgments

We are grateful to present and former members of the Hinton laboratory for helpful discussions, particularly Xiaojun Zhu and Carsten Kröger, and to Paul Loughnane for his technical assistance. We thank John Kenny for his expertise in library preparation for Illumina sequencing, and thank the Centre for Genomic Research at the University of Liverpool for the use of the Covaris equipment. We appreciated our helpful discussions with Duy Phan (University of Queensland) concerning the design and analysis of the transposon insertion experiments.

## Author Contributions

**Conceptualization:** Rocío Canals, Michael Ibba, Jay C. D. Hinton.

**Data curation:** Roy R. Chaudhuri, Natalia Quinones-Olvera.

**Formal analysis:** Rocío Canals, Roy R. Chaudhuri, Siân V. Owen, Natalia Quinones-Olvera.

**Funding acquisition:** Rocío Canals, Michael Ibba, Jay C. D. Hinton.

**Investigation:** Rocío Canals, Roy R. Chaudhuri, Rebecca E. Steiner, Siân V. Owen, Natalia Quinones-Olvera.

**Methodology:** Rocío Canals, Roy R. Chaudhuri, Rebecca E. Steiner, Siân V. Owen, Natalia Quinones-Olvera, Michael Ibba.

**Project administration:** Rocío Canals.

**Resources:** Melita A. Gordon, Michael Baym, Michael Ibba, Jay C. D. Hinton.

**Software:** Roy R. Chaudhuri.

**Supervision:** Rocío Canals, Siân V. Owen, Melita A. Gordon, Michael Baym, Michael Ibba, Jay C. D. Hinton.

**Validation:** Rocío Canals, Rebecca E. Steiner.

**Visualization:** Rocío Canals, Roy R. Chaudhuri, Rebecca E. Steiner, Siân V. Owen, Natalia Quinones-Olvera.

**Writing – original draft:** Rocío Canals, Jay C. D. Hinton.

**Writing – review & editing:** Rocío Canals, Roy R. Chaudhuri, Rebecca E. Steiner, Siân V. Owen, Natalia Quinones-Olvera, Michael Baym, Michael Ibba, Jay C. D. Hinton.

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
