## [Decision Letter · Decision Letter 0]

27 Jul 2019

Dear Dr. Canals:

Thank you very much for submitting your manuscript "The fitness landscape of the African Salmonella Typhimurium ST313 strain D23580 reveals unique properties of the pBT1 plasmid" (PPATHOGENS-D-19-01144) for review by PLOS Pathogens. Your manuscript was fully evaluated at the editorial level and by independent peer reviewers. The reviewers appreciated the attention to an important topic but identified some aspects of the manuscript that should be improved.

We therefore ask you to modify the manuscript according to the review recommendations before we can consider your manuscript for acceptance. Your revisions should address the specific points made by each reviewer.

(1) A letter containing a detailed list of your responses to the review comments and a description of the changes you have made in the manuscript. Please note while forming your response, if your article is accepted, you may have the opportunity to make the peer review history publicly available. The record will include editor decision letters (with reviews) and your responses to reviewer comments. If eligible, we will contact you to opt in or out.

(2) Two versions of the manuscript: one with either highlights or tracked changes denoting where the text has been changed; the other a clean version (uploaded as the manuscript file).

We hope to receive your revised manuscript within 60 days or less. If you anticipate any delay in its return, we ask that you let us know the expected resubmission date by replying to this email.

[LINK]

Sincerely,

Andreas J Baumler

Associate Editor

PLOS Pathogens

Renée Tsolis

Section Editor

PLOS Pathogens

Kasturi Haldar

Editor-in-Chief

PLOS Pathogens

orcid.org/0000-0001-5065-158X

Grant McFadden

Editor-in-Chief

PLOS Pathogens

orcid.org/0000-0002-2556-3526

Reviewer's Responses to Questions

**Part I - Summary**

Reviewer #1: The molecular basis of pathogen adaptation to different environments is still not completely understood. The Hinton lab has made major contributions to this field and established powerful web based platforms to make this data easily accessible to the field. In the current paper, they have focused on a strain of ST313, which currently dominates the worldwide Typhimurium infections. Specifically, they performed a comparative high saturation INSEQ screen to identify genes required for growth of D23580 in LB, in a SPI2 inducing minimal medium and in macrophages. The data iscarefully evaluated and the pros and cons of the approach are well discussed.

The data provide an exciting complementary dataset not only for comparison to similar data on the model strains commonly used in the field, but also for enriching the Hinton lab's spectacular online resources on Salmonella gene function. This will be of exceptionally high value. On top of this general importance, the screen identified an intriguing case of functional replacement of an otherwise essential chromosomal gene by a plasmid-encoded aatRNA synthetase, cysSpBT1. The biochemical characteristics of the chromosomal and the plasmid encoded enzyme were compared and my point towards a role in the stringent response. This would be an intriguing starting point for future studies.

The paper is very well written, and will serve as a valuable resource for functional information for the field. I only have one minor comment.

Reviewer #2: In this study, Canals et al perform a genetic screen to determine unique fitness determinants of an iNTS Salmonella isolate during growth in cultured macrophages and in laboratory conditions. A few “hits” were validated in separate assays, for example arginine biosynthesis is required for optimal replication in macrophages in iNTS and gastroenteritis-associated Salmonella Typhimurium. Overall, the authors found that the genetic requirements for growth under these conditions are similar to that to other, gastroenteritis-associated and typhoidal Salmonella strains. The authors found that many plasmid-encoded genes appeared to be essential for efficient replication of iNTS inside cultured macrophages. A plasmid-encoded aminoacyl tRNA synthetase, CysRS, had increased activity compared to the chromosomally-encoded paralogue but was less stable in vitro. Through bioinformatics, other plasmid encoded aminoacyl tRNA synthetases are identified.

iNTS is a significant public health problem in developing countries, yet the molecular mechanisms of pathogenesis are incompletely understood. This study touches on an important topic as compares the genetic requirements for growth in iNTS, gastroenteritis-associated strains, and typhoidal strain (done previously by the Hinton group). The data are solid; the effort by the authors in generating and analyzing the data, especially cross-referencing other studies, are very much appreciated. The authors deserve a lot of credit for carefully discussing the limitations of their study. I have two comments:

1. The study revealed little differences between iNTS and gastroenteritis-associated strains. Since only a small number of conditions were tested, only limited conclusions can be drawn from this comparison. If the conclusion of the study was that there are little differences in the genetic requirements for iNTS and gastroenteritis-associated strains (favored by the authors), then this analysis is suffers from not being comprehensive as only a few conditions were tested; conversely, if the main conclusion of the study was that there are indeed differences with relevance to iNTS pathogenesis, then this study did not find provide experimental evidence to support this conclusion.

2. Very little new biology, beyond what one would expect already (the authors do a great job at incorporating current knowledge into their analysis), is revealed. The authors argue that differences in the activity and thermal stability of CysRS may allow the organism to cope better with stress, for example by inducing the stringent response. This is plausible, but not shown experimentally and it is not clear whether this phenomenon is relevant for iNTS pathogenesis.

Reviewer #3: Canals et al present a transposon-insertion sequencing study of the S. Typhimurium ST313 strain D23580. The study examines the fitness of D23580 transposon insertion mutants during in vitro growth and during infection of a murine macrophage cell line, and identifies a range of required gene products and intergenic regions needed for growth in a given context. Altogether, this is a well-written manuscript that provides substantial insight into an important pathogen's biology, highlights the importance of a very interesting class of plasmid-encoded required genes, and in general should serve as a great resource for the community to leverage and perform followup studies.

**Part II – Major Issues: Key Experiments Required for Acceptance**

Reviewer #1: no additional experiments required.

Reviewer #2: (No Response)

Reviewer #3: 1. Considering the employed transposon's potential for driving the overexpression of genes downstream of the aph gene's promoter, the potential for interrupting downstream gene transcription in operons when the transposon is inserted in the 'antisense'-orientation (relative to the aph promoter), and the high level of genome saturation achieved in the study (an insertion every 10nt on average), the authors should (and should be able to) include an analysis of transposon insertion orientation bias within genes and intergenic regions, and correlate each entity's bias in the input and output pools (e.g., "gene X: Input='no bias'; Output='biased towards antisense insertions'"). One can envision a number of scenarios where polarity stemming from one/both/neither insertion orientations could improve or diminish the organism's fitness in the employed assays. Although the authors touch on aspects of polarity in the text, for example when rationalizing why some genes exhibit low insertion density, a comprehensive accounting of insertion orientation bias would provide further insight into (and confidence in) the TIS results, as well as potentially help to further rationalize discrepancies concerning necessity/dispensability of genes.

**Part III – Minor Issues: Editorial and Data Presentation Modifications**

Reviewer #1: Line 176: the authors should comment whether this observation is truly novel. Have other publications previously overlooked that essential genes are often highly expressed?

Reviewer #2: (No Response)

Reviewer #3: 1. On Line 277, please change "reveals" to "suggests" (akin to what is written in Lines 37 and 496). Bacterial mutants which are defective in the production of a given host-disarming gene product are, conceivably, subject to rescue by other co-infecting mutants in the pool (i.e., the rest of the pool is still competent for the host-disarming function). Individually screening the mutants, albeit beyond the scope of this manuscript, would have greater resolution in identifying novel virulence factors related to intra-macrophage survival, and thus such novel virulence factors could still exist in D23580.

PLOS authors have the option to publish the peer review history of their article (what does this mean?). If published, this will include your full peer review and any attached files.

Reviewer #1: No

Reviewer #2: No

Reviewer #3: No

---

## [Editor Report · Decision Letter 1]

30 Aug 2019

Dear Dr. Canals,

We are pleased to inform that your manuscript, "The fitness landscape of the African Salmonella Typhimurium ST313 strain D23580 reveals unique properties of the pBT1 plasmid", has been editorially accepted for publication at PLOS Pathogens. 

Before your manuscript can be formally accepted and sent to production, you will need to complete our formatting changes, which you will receive by email within a week. Please note that your manuscript will not be scheduled for publication until you have made the required changes.

IMPORTANT NOTES

(1) Please note, once your paper is accepted, an uncorrected proof of your manuscript will be published online ahead of the final version, unless you’ve already opted out via the online submission form. If, for any reason, you do not want an earlier version of your manuscript published online or are unsure if you have already indicated as such, please let the journal staff know immediately at plospathogens@plos.org.

(2) Copyediting and Proofreading: The corresponding author will receive a typeset proof for review, to ensure errors have not been introduced during production. Please review the PDF proof of your manuscript carefully, as this is the last chance to correct any errors. Please note that major changes, or those which affect the scientific understanding of the work, will likely cause delays to the publication date of your manuscript. 

(3) Appropriate Figure Files: Please remove all name and figure # text from your figure files. Please also take this time to check that your figures are of high resolution, which will improve the readbility of your figures and help expedite your manuscript's publication. Please note that figures must have been originally created at 300dpi or higher. Do not manually increase the resolution of your files. For instructions on how to properly obtain high quality images, please review our Figure Guidelines, with examples at: http://journals.plos.org/plospathogens/s/figures.

(4) Striking Image: Please upload a striking still image to accompany your article if one is available (you can include a new image or an existing one from within your manuscript). Should your paper be accepted, this image will be considered for our monthly issue image and may also appear on our website to feature your article. Please upload this as a separate file, selecting "striking image" as the file type upon upload. Please also include a separate "Other" file with a caption, including credits and any potential copyright information. Please do not include the caption in the main article file. If your image is from someone other than yourself, please ensure that the artist has read and agreed to the terms and conditions of the Creative Commons Attribution License at http://journals.plos.org/plospathogens/s/content-license. Please note that PLOS cannot publish copyrighted images.

(5) Press Release or Related Media: If your institution or institutions have a press office, please notify them about your upcoming paper at this point, to enable them to help maximize its impact. If they will be preparing press materials for this manuscript, please inform our press team in advance at plospathogens@plos.org as soon as possible. We ask that you contact us within one week to plan ahead of our fast Production schedule. If you need to know your paper's publication date for related media purposes, you must coordinate with our press team, and your manuscript will remain under a strict press embargo until the publication date and time. This means an early version of your manuscript will not be published ahead of your final version. 

(6)  PLOS requires an ORCID iD for all corresponding authors on papers submitted after December 6th, 2016. Please ensure that you have an ORCID iD and that it is validated in Editorial Manager.  To do this, go to ‘Update my Information’ (in the upper left-hand corner of the main menu), and click on the Fetch/Validate link next to the ORCID field.  This will take you to the ORCID site and allow you to create a new iD or authenticate a pre-existing iD in Editorial Manager

(7) Update your Profile Information: Now that your manuscript has been provisionally accepted, please log into Editorial Manager and update your profile, if needed. Go to https://www.editorialmanager.com/ppathogens, log in, and click on the "Update My Information" link at the top of the page. Please update your user information to ensure an efficient production and billing process. 

(8) LaTeX users only: Our staff will ask you to upload a TEX file in addition to the PDF before the paper can be sent to typesetting, so please carefully review our Latex Guidelines http://journals.plos.org/plospathogens/s/latex in the meantime.

(9) If you have associated protocols in protocols.io, please ensure that you make them public before publication to guarantee immediate access to the methodological details.

Best regards,

Andreas J Baumler

Associate Editor

PLOS Pathogens

Renée Tsolis

Section Editor

PLOS Pathogens

Kasturi Haldar

Editor-in-Chief

PLOS Pathogens

orcid.org/0000-0001-5065-158X

Grant McFadden

Editor-in-Chief

PLOS Pathogens

orcid.org/0000-0002-2556-3526
---

## [Editor Report · Acceptance letter]

20 Sep 2019

Dear Dr. Canals,

We are delighted to inform you that your manuscript, "The fitness landscape of the African Salmonella Typhimurium ST313 strain D23580 reveals unique properties of the pBT1 plasmid," has been formally accepted for publication in PLOS Pathogens.

Best regards,

Kasturi Haldar

Editor-in-Chief

PLOS Pathogens

orcid.org/0000-0001-5065-158X

Grant McFadden

Editor-in-Chief

PLOS Pathogens

orcid.org/0000-0002-2556-3526